# Diagnosing drivers of PM$_{2.5}$ simulation biases in China from meteorology, chemical composition, and emission sources using an efficient machine learning method

Shuai Wang[1], Mengyuan Zhang[1], Yueqi Gao[1], Peng Wang[2,3], Qingyan Fu[4], Hongliang Zhang[1,3,5]

[1]Department of Environmental Science and Engineering, Fudan University, Shanghai 200438, China

[2]Department of Atmospheric and Oceanic Sciences, and Institute of Atmospheric Sciences, Fudan University, Shanghai, 200438, China

[3]IRDR ICoE on Risk Interconnectivity and Governance on Weather/Climate Extremes Impact and Public Health, Fudan University, Shanghai, China;

[4]Shanghai Environmental Monitoring Center, Shanghai 200235, China

[5]Institute of Eco-Chongming (IEC), Shanghai 200062, China

*Correspondence to*: Hongliang Zhang (zhanghl@fudan.edu.cn)

**Abstract.** Chemical transport models (CTMs) are widely used for air pollution modeling, which suffer from significant biases due to uncertainties in simplified parameterization, meteorological fields, and emission inventories. Accurate diagnosis of simulation biases is critical for improvement of models, interpretation of results, and management of air quality, especially for the simulation of fine particulate matter (PM$_{2.5}$). In this study, an efficient method with fast speed and low requirement of computational resources based on the tree-based machine learning (ML) method, the Light Gradient Boosting Machine (LightGBM), was designed to diagnose CTMs simulation biases. The drivers of the Community Multiscale Air Quality (CMAQ) model biases compared to observations in simulating PM$_{2.5}$ concentrations from three perspectives of meteorology, chemical composition, and emission sources. The source-oriented CMAQ was used to diagnose the influences of different emission sources on PM$_{2.5}$ biases. The model can capture the complex relationship between input variables and simulation bias well, meteorology, PM$_{2.5}$ components, and source sectors can partially explain the simulation bias. The CMAQ model underestimates PM$_{2.5}$ by -19.25 to -2.66 μg/m$^3$ in 2019, especially in winter and spring and high PM$_{2.5}$ events. Secondary organic components showed the largest contribution to PM$_{2.5}$ simulation bias for different regions and seasons (13.8 - 22.6%) among components. Relative humidity, cloud cover, and soil surface moisture were the main meteorological factors contributing to PM$_{2.5}$ bias in the North China Plain, Pearl River Delta, and Northwestern, respectively. Both primary and secondary inorganic components from residential sources showed the largest contribution to this bias (12.05 % and 12.78 %), implying large uncertainties in this sector. The ML-based methods provide valuable complements to traditional mechanism-based methods for model improvement, with high efficiency and low reliance on prior information.

## 1 Introduction

Fine particulate matter (PM$_{2.5}$) is a complex mixture of primary particulate matter (PPM) and secondary inorganic/organic components (SIA/SOA), with adverse effects on public health and ecosystems. Ambient levels of PM$_{2.5}$ are influenced by meteorological conditions, emission from different sources, and atmospheric chemical processes (Organization, 2021; Xiao et al., 2021a; Yang et al., 2016; Liu et al., 2021b; Zhai et al., 2019). China has experienced severe PM$_{2.5}$ pollution over the past two decades (Bai et al., 2022; Liang et al., 2020a). For effective air quality management, accurate PM$_{2.5}$ modeling is essential. Chemical transport models (CTMs) like the Community Multiscale Air Quality (CMAQ) model, have been widely developed and applied to PM$_{2.5}$ simulations through atmospheric processes of dispersion, deposition, and chemical reactions (Qiao et al., 2018; Wang et al., 2021; Hu et al., 2017a). Application of CTMs simulations is often limited by the biases due to uncertainties of simplified parameterization, meteorological prediction, emission inventories, and initial and boundary conditions

(Binkowski and Roselle, 2003; Hu et al., 2014; Hu et al., 2016; Wang et al., 2023b; Wang et al., 2021). Thus, it is essential to diagnose specific sources of simulation biases according to specific model applications, including grid resolution, parameterization, mechanisms, meteorological inputs, and emission inventories.

Traditional bias diagnosis approaches for CTM models usually rely on empirical and prior assumptions with extensive sensitivity testing and high demands on computational resources, such as Monte Carlo methods or Latin hypercube sampling

(Beekmann and Derognat, 2003; Hanna et al., 2005; Aleksankina et al., 2019). Recently machine learning (ML) methods, like Random Forest and eXtreme Gradient Boosting (XGBoost), have been widely used in environmental science researches due to their simple structure, fast speed, and ability to deal with no-linear relationships (Liu et al., 2022). Many studies used ML to predict air pollutant concentrations like $PM_{2.5}$ and ozone (Wei et al., 2021a; Sun et al., 2021; Zhu et al., 2022; Bai et al., 2022), improve the accuracy of CTMs simulations (Wang et al., 2023b; Wei et al., 2020), and explain the prediction results

using interpretable ML techniques (Hou et al., 2022; Li et al., 2023; Stirnberg et al., 2021). To date, few studies have used ML to diagnose the simulation bias of CTMs. A study has shown the potential of machine learning in explaining the simulation bias of ozone (Ye et al., 2022). However, as a complex multi-phase mixture, it is still challenging to diagnose biases in $PM_{2.5}$ simulations using ML methods (Liu and Xing, 2022). Moreover, given the significant impact of emissions, it is instructive to diagnose CTMs biases of $PM_{2.5}$ based on a source-appointment perspective.

In this study, we use LightGBM model, an efficient ensemble ML approach, to diagnose the drivers of CMAQ biases in simulating $PM_{2.5}$ concentrations. A source-oriented version of CMAQ is used to track sectoral source contributions to $PM_{2.5}$. Model biases are diagnosed from multiple perspectives, including meteorology, chemical components, and emission sources.

## 2 Materials and methods

### 2.1 Surface $PM_{2.5}$ observations

This study specifically targets the year of 2019 due to the extensive availability of observational data, the reliability of emission inventories, and the absence of COVID-19-related disruptions. Hourly $PM_{2.5}$ observations for 2019 are collected from the China National Environmental Monitoring Centre (CNEMC, http://www.cnemc.cn/). The daily observations data <0.1 % quantile and >99.9 % quantile, $PM_{2.5}$ exceeds $PM_{10}$, and days with less than 20 valid hourly records are excluded. For observation sites located on the same CMAQ simulation grid (36 km × 36 km), average $PM_{2.5}$ concentrations of these sites

were calculated to compare with CMAQ simulation. Approximately 350,000 observations, which met the quality control criteria, were selected from the entire time series data points collected from various monitoring stations. The distribution of observation sites (about 1200) is shown in Figure S1. The stations are unevenly distributed, with dense stations in eastern populated areas and sparse stations in western Xinjiang and Tibet. Analysis has been carried out on several haze-polluted regions and the whole country (Figure S1), including Beijing-Tianjin-Hebei (BTH); the Yangtze River Delta (YRD); the Pearl

River Delta (PRD); the Sichuan Basin (SCB); and Northwestern China (NWCHN).

### 2.2 CMAQ simulation

The CMAQ simulation (36 km×36 km) was carried out to simulate $PM_{2.5}$ components in mainland China and surrounding regions in 2019. The list of $PM_{2.5}$ components simulated by CMAQ is shown in Table A1. The Weather Research & Forecasting Model (WRF v4.2) was used to generate meteorological fields driven by the National Centers for Environmental Prediction

(NCEP) FNL Operational Model Global Tropospheric Analyses dataset (http://rda.ucar.edu/datasets/ds083.2/) (Commerce, 2000). Several meteorological factors (Table A1) that are highly relevant to aerosol concentrations are selected for ML model building (Xiao et al., 2021b; Chen et al., 2020b; Meng et al., 2019). The CMAQ v5.0.2 with a modified SAPRC-11 photochemical mechanism and AERO6 aerosol module was applied for aerosol simulations (Carter and Heo, 2013; Ying et al., 2015; Binkowski and Roselle, 2003). The Multi-resolution Emission Inventory for China (MEIC) was used as

anthropogenic emission (http://meicmodel.org/), and the Model for Emissions of Gases and Aerosols from Nature (MEGAN) version 2.1 was used to generate biogenic emissions (Guenther et al., 2012; Guenther et al., 2006). The Fire INventory from NCAR (FINN) based on satellite was used to generate open burning emissions (Wiedinmyer et al., 2011).

The source apportionment method was used to quantify the contributions of industry, energy, residential, transportation, agriculture, open burning, and biogenic sources to PPM and SIA through a modified version of CMAQ (Zhang et al., 2012; Ma et al., 2021; Qiao et al., 2018). PPM from different source sectors are tracked by non-reactive tracers ($10^{-5}$ of the PPM emission rates), and source-specific PPM concentrations are then calculated by multiplying the tracer with $10^5$. The contributions of source sectors to SIA are quantified using specific reactive tagged tracers. Specifically, $NO_x$, $SO_2$, and $NH_3$ from different sources were tracked separately through a series of chemical and physical processes involved in SIA formation. The source of SOA was not traced due to the complex and currently imperfect mechanism of SOA formation and the high uncertainties in the precursor VOCs emissions (Liu et al., 2021b; Hu et al., 2017b). Details of source apportionment can be found in previous studies (Zhang et al., 2012; Ma et al., 2021; Qiao et al., 2018; Ying et al., 2014). The contributions of source sectors to SOA were not tracked due to insufficient knowledge of its precursors and incomplete formation mechanisms (Yang et al., 2019; Carlton et al., 2007; Zhang et al., 2011).

### 2.3 Machine learning method

Tree-based ML models typically outperform deep learning approaches in tabular data (e.g., air pollutant observation datasets), and thus have been widely developed (Grinsztajn et al., 2022). Wei et al. (2021a) compared several models when reconstructing $PM_{2.5}$ data records in China and found that the tree model showed superior performance. The LightGBM model is an optimized Gradient Boosting Decision Tree (GBDT) (Ke et al., 2017), and has shown accurate performance in many fields (Wei et al., 2021b; Yan et al., 2021; Sun et al., 2020; Liang et al., 2020b). Compared to XGBoost, a widely used GBDT, LightGBM uses Histogram's decision tree algorithm along with Gradient-based One-Side Sampling (GOSS), which saves memory and computation time (Ke et al., 2017). Three tree-based models, Random Forest, XGBoost, and LightGBM, were compared in our previous study (Wang et al., 2023a). Using the same input data and hyperparameters, LightGBM is as accurate as XGBoost, but faster and less overfitting (the difference in accuracy between training and testing). Besides, Multiple colinearities between features such as pollutant concentrations and meteorological factors can greatly affect the performance of traditional linear models. When independent variables are correlated, changes in one variable are associated with changes in the other, making it difficult for the model to independently estimate the relationship between each independent and dependent variable. However, these collinearities do not affect the performance of tree-based models like Random Forest and LightGBM, because they do not require the assumption of feature independence (Belgiu and Drăguţ, 2016; Chen et al., 2016; Ke et al., 2017). So, the lightGBM model was used to diagnose $PM_{2.5}$ simulation biases in this study. Two metrics were calculated to evaluate model performance, including the coefficient of determination ($R^2$) and the root mean square error (RMSE) (Wei et al., 2020).

$$R^2 = 1 - \frac{\Sigma_i(y_i - f_i)^2}{\Sigma_i(y_i - \hat{y})^2} \tag{1}$$

$$\text{RMSE} = \sqrt{\frac{1}{n}\Sigma_{i=1}^{n}(y_i - f_i)^2} \tag{2}$$

Cross-validation (5-fold) combined with RMSE was used to select hyperparameters. The dataset was randomly divided into five parts, one was taken in turn as the test set, and the rest was used for training, which was repeated five times, and the average test RMSE was calculated. Looping to increase model complexity, ending the loop and returning to the hyperparameters when the model test RMSE does not decrease significantly (< 0.01) or the gap between training and test RMSE increases significantly (< 0.05). The separate test sets (not involved in the training and CV hyperparameter selection process) were divided by randomly sampling 20% of the data from all stations in the region of interest.

The target variable was defined as the difference between observed and simulated daily $PM_{2.5}$ concentrations, and the key

contributors to the simulation bias were identified through the relative importance (calculated by gain) of the input features (Ye et al., 2022; Loyola-González et al., 2023). Three categories of input variables were designed to separately fit the simulation biases: meteorological factors, chemical components, and emission sources. Meteorological factors, including wind speed, wind direction, temperature, humidity, surface pressure, cloud fraction, and boundary layer height, are used to investigate the impact of meteorology on the CMAQ simulation biases. The components of $PM_{2.5}$ are divided into SIA ( sulfate, nitrate, ammonium), primary/secondary organic aerosols (POA/SOA), elemental carbon (EC), and other components. The contributions to the simulation bias were quantified using seven sectoral sources: industry, energy, residential, transportation, agriculture, open burning, and biogenic emissions.

## 3 Results and discussion

### 3.1 Observation and simulation of $PM_{2.5}$

Figure 1a shows the time series of observed and simulated daily surface $PM_{2.5}$ concentrations in China and five regions (BTH, YRD, PRD, SCB, and NWCHN) over 2019. Observed $PM_{2.5}$ concentrations were highest in BTH (51.172 μg/m$^3$) and lowest in PRD (28.273 μg/m$^3$). The CMAQ model underestimates $PM_{2.5}$ concentrations of -8.59 μg/m$^3$, -2.66 μg/m$^3$, -6.21 μg/m$^3$, and -19.25 μg/m$^3$ in the BTH, YRD, PRD, and NWCHN, respectively (Figure 1b). Moreover, the underestimation occurred mainly in winter and spring (Figure 1c), as well as high $PM_{2.5}$ events (Figure 1d) (Hu et al., 2016; Huang et al., 2017).

Table A2 shows the validation of CMAQ simulations against observations in different regions. Four indicators (MNB: mean normalized bias; MNE: mean normalized error; MFB: mean fractional bias; MFE: mean fractional error) were used to systematically evaluate the performance of the CMAQ simulations. The $PM_{2.5}$ simulations in the BTH, YRD, and PRD regions were in better agreement with observations, with average MNB of -0.08, -0.07, and -0.08 respectively (within the standard of 0.66). The $PM_{2.5}$ simulations in SCB and NWCHN regions show large biases with MNB of 0.46 and -0.42 respectively. The differences of CMAQ performance between regions can be attributed to multiple factors including emission inventories, dominant mechanisms of $PM_{2.5}$ generation, topographic, and meteorology conditions (Ma et al., 2021; Xue et al., 2019; Hu et al., 2014).

Annual and monthly mean $PM_{2.5}$ components (SIA, POA, SOA, EC, and other components) were calculated for China and five key regions (Figure 2). $PM_{2.5}$ and its components show similar spatial distribution, with high concentrations occurring in the eastern regions (SCB, BTH, and central YRD). SOA showed high concentrations in summer over China (6.80 μg/m$^3$), which could be related to enhanced solar radiation and atmospheric oxidation capacity in summer (precursors of SOA such as isoprene are highly dependent on temperature and light) (Yang et al., 2019; Chen et al., 2020a; Liu et al., 2021b). Nitrate and POA were the dominant components in winter (10.14 μg/m$^3$ and 9.11 μg/m$^3$ respectively). In BTH and SCB, POA accounts for a higher proportion than nitrate in winter, whereas nitrate has a higher proportion in YRD. Nitrate showed higher concentration than sulfate in most regions and seasons due to the implementation of coal combustion control policies (Shang et al., 2021; Liu et al., 2021b; Xu et al., 2019).

The results of the $PM_{2.5}$ sectoral source appointment (Figure 3 and Figure S2) show that industries and residential sources were the main contributors to daily $PM_{2.5}$ concentrations for all regions and seasons, with seasonal fractional contributions of 25.31 - 31.92 % and 11.13 - 30.64 %, respectively). The seasonal average fractional contributions from energy, transportation, and agricultural $NH_3$ in the whole China were 3.26 - 5.67%, 6.82 - 11.26 %, and 7.50 - 8.67 %, respectively. The contributions from biogenic source were negligible in all regions and seasons (< 1 %). In contrast to the contributions from energy, transportation, industrial, and agricultural sources, significant seasonal variations occurred from residential source in all five regions, with high contribution in winter (17.60 - 30.90 %) and low contribution in summer (5.53 - 16.46 %).

As the secondary component makes up a large proportion of the total $PM_{2.5}$, the source sectors of SIA were analyzed for five regions (Figure S2). High concentrations of SIA were found in winter (12.36 - 34.08 μg/m$^3$), with large contribution from

industrial, agricultural, and transportation sources (31.49 - 36.41 %, 20.40 -22.40 %, and 19.77 - 22.46 %). The low contribution of the residential sector to secondary $PM_{2.5}$ but the high contribution to total $PM_{2.5}$ indicates that most residential emission sources emit PPM directly, with a small fraction of secondary generation. The contributions from biogenic and open burning sectors to SIA were relatively low in all regions and seasons (< 10 %).

## 3.2 Drivers of $PM_{2.5}$ simulation bias

The ML models were trained separately using meteorology, $PM_{2.5}$ components, and source sectors for different regions and seasons, and separate test sets were used to evaluate the model performance (Figure 4). All three feature combinations can partially explain the simulation bias. The mean test $R^2$ for meteorology, $PM_{2.5}$ components, and source sectors were 0.64, 0.52, and 0.50, respectively, and the RMSE was 17.41, 19.82, and 19.56 $\mu g/m^3$, respectively. The model performed better in summer than in winter. This may be attributed to the high simulation biases in winter due to severe $PM_{2.5}$ pollution and complex causes, while $PM_{2.5}$ pollution in summer is lighter with lower CMAQ simulation bias.

Using $PM_{2.5}$ components as input features to fit the total simulation bias enables the identification of components with large simulation bias. Among the $PM_{2.5}$ components (Figure S4), SOA showed the largest contribution to $PM_{2.5}$ simulation bias for different regions and seasons (13.8 - 22.6%), which is consistent with previous studies (Liu et al., 2021b; Yang et al., 2019; Fry et al., 2014). The inorganic aerosols (e.g. sulfates) are produced mainly by chemical pathways, while SOA is produced by the condensation of numerous partially oxidized gases and is therefore influenced by complex precursor concentrations and multi-stage oxidation processes. The incomplete description of SOA formation pathways in CTMs models (simplified SOA parameterization) leads to significant differences between simulations and observations (Carlton et al., 2007; Zhang et al., 2018; Yang et al., 2019). In addition, biogenic emissions play an important role in SOA formation, with biogenic SOA accounting for more than 70% of total SOA in China during summer (Hu et al., 2017b; Wu et al., 2020), so uncertainties in biogenic emissions can further contribute to uncertainties in SOA. Nitrate showed a large contribution to $PM_{2.5}$ simulation bias in winter at BTH, which is consistent with the previous study (Liu and Xing, 2022). Nitrate contribution to simulation bias further implies the inaccuracy of nitrate simulations, which can relate to the imperfect pathways of nitrate production (e.g., non-homogeneous oxidation) in SAPRC11 (that we used) and the uncertainties of nitrate precursor emission inventories in winter (Xu et al., 2019; Zhang et al., 2018; Carter and Heo, 2013).

The contribution of meteorological factors to the simulation bias varies across regions and seasons (Figure 5). In the BTH region, surface pressure and relative humidity contribute the most to the simulation bias. In the PRD region, relative humidity, cloud cover, and wind direction were the main contributors in all four seasons.

Humidity positively or negatively influences $PM_{2.5}$ concentrations through several mechanisms. By enhancing $PM_{2.5}$ hygroscopic increase, promoting the secondary formation, and facilitating the gas-to-particle partitioning, high humidity positively influences $PM_{2.5}$ concentrations and contributes significantly to haze pollution (Chen et al., 2020b; Cheng et al., 2015; Zhang et al., 2011). The contribution of humidity to CMAQ simulation biases can partly attributed to the uncertainties of WRF simulation. The mean RMSE of relative humidity from WRF simulations versus observations was 20.38% in this study (Table A3). In addition, imperfections in the mechanism of humidity-promoted secondary particle formation (e.g., non-homogeneous catalysis of SOA) can also lead to simulation biases (Zhang et al., 2011; Liu et al., 2021b). Atmospheric pressure indirectly influences $PM_{2.5}$ concentrations through other meteorological factors (e.g., humidity, wind, etc.). High-pressure systems are connected to stationary weather, which is unfavorable for $PM_{2.5}$ dispersion. On the other hand, low pressure is usually accompanied by high humidity, influencing $PM_{2.5}$ nucleation, condensation, and coagulation, leading to increased $PM_{2.5}$ concentrations (Chen et al., 2020b). Therefore, the influence of atmospheric pressure on the CMAQ simulation biases in the BTH region may be attributed to the uncertainties of meteorological field (Bei et al., 2017; Zhang et al., 2015). The contribution of wind direction in YRD may also related to the uncertainties of WRF simulation (mean RMSE: 90.39 °). Aerosols have feedback on meteorology (Qu et al., 2021). In addition to directly changing the radiation received by the earth

through scattering and absorbing (direct radiation effect), $PM_{2.5}$ can also influence radiation through aerosol-cloud interactions (indirect radiation effect) (Zhao et al., 2017; Yang et al., 2016). Moreover, $PM_{2.5}$ can act as cloud condensation and nucleation sites, contributing to clouds' microphysical development and precipitation formation process (Wang et al., 2020). However, the aerosol-to-meteorological feedback mechanism is missing in CMAQ used in this study. A previous study showed the dominant role of cloud chemistry in aerosol-cloud interactions in southern China (Zhao et al., 2017). Therefore, the influence of cloud cover on simulation biases in YRD can attributed to the lack of aerosol feedback mechanism.

In the NWCHN region, soil surface moisture and stomatal conductance contributed significantly to the simulation bias. These factors can be associated with ground-level sand rise and dust emission (Liu et al., 2021c). Underestimation of dust aerosol in NWCHN can be attributed to emission (both natural and anthropogenic sources), and an accurate emission inventory (empirical- or physical-based numerical models) should be established in Northwest China by detailed activity data and emission factors (Hu et al., 2016; Liu et al., 2021a). In addition, the parameterization and mechanism for dust aerosol simulation in CMAQ should be further examined and improved.

Dry and wet days were divided to analyze the influence of humidity on the simulation biases (Table A4). In most areas of China, CMAQ underestimates $PM_{2.5}$ more severely on dry days than on wet days, with larger absolute biases (-14.56 µg/m³, -7.09 µg/m³, -7.11 µg/m³, and -27.87 µg/m³ in spring, summer, autumn, and winter respectively). In dry days, BTH showed severe underestimation in winter (-22.86 µg/m³), while PRD showed large simulation bias in spring (-21.55 µg/m³). Severe underestimation of $PM_{2.5}$ was observed in both wet and dry days at NWCHN. These underestimates of $PM_{2.5}$ in dry days can related to the dry deposition process. Dry deposition is a critical but highly uncertain sink for aerosols, which depends on the chemical and physical properties of aerosols, and is influenced by land surface properties and meteorological conditions (Shu et al., 2022). Different land-use types (e.g., vegetation, deserts, and snow) possess markedly different capacities to capture particulate matter. The CMAQ model in this study used the dry deposition scheme PR11 from Pleim and Ran (Pleim and Ran, 2011). This study shows that the PR11 scheme underestimates $PM_{2.5}$ concentrations in China. Recent studies in the United States also showed an underestimation of $PM_{10}$ concentrations (Shu et al., 2022). Therefore, it is necessary to further develop and optimize the dry deposition scheme, especially for $PM_{2.5}$. $PM_{2.5}$ underestimation in wet days may be attributed to the biases of wet deposition and secondary organic aerosol formation under high humidity conditions (Wu et al., 2018; Ryu and Min, 2022; Liu et al., 2021b; Zhang et al., 2011).

Source sector contributions of PPM and SIA (obtained from the source-oriented CMAQ) were used to build the model and diagnose the influences of different emissions sources on $PM_{2.5}$ simulation biases (Figure 6). The PPM and SIA from residential showed the largest contribution (12.05 % and 12.78 %) to $PM_{2.5}$ simulation bias. The same conclusion was obtained when building a model with total $PM_{2.5}$ concentrations from different source sectors (Table A5). $PM_{2.5}$ from residential emissions is the main contributor to the CMAQ simulation bias, accounting for 20.2% of the total bias.

In China, the residential sector consumed fossil fuels (coal, oil, and natural gas) and biofuels (wood and crop straw) with low combustion efficiency for cooking and heating and emitted large amounts of air pollutants (Li et al., 2017). However, due to the lack of reliable data (locally accurate emission factor and fuel consumption data), the residential sector has been recognized as a major uncertainty source in anthropogenic emission inventories (Liu et al., 2021d; Shen et al., 2021), which is consistent with the results identified by machine learning in this study. Therefore, developing an accurate residential sector emissions inventory is essential for accurate $PM_{2.5}$ modeling, which requires reliable data of fuel consumption and emission factors based on fuel type, fuel characteristics, and combustion conditions (Liu et al., 2021d).

### 3.3 Comparisons and uncertainties

Huang et al. (2019) used a new reduced-form model coupled with a higher-order decoupled direct method and stochastic response surface model to identify sources of uncertainty in CMAQ simulations. An analysis of the PRD of China in the spring of 2013 revealed a systematic underestimation of SOA and identified wind speed and primary $PM_{2.5}$ emissions as key sources

of uncertainties in $PM_{2.5}$ simulations, which is consistent with the results identified using LightGBM in this study. Aleksankina et al. (2019) identified $PM_{2.5}$ simulation bias in Europe using optimised Latin hypercube sampling and also demonstrated the important impact of primary emissions on $PM_{2.5}$ simulation uncertainties. Liu and Xing (2022) used a fully connected neural network to identify $PM_{2.5}$ simulation biases and found that $NO_2$ is the main contributor in BTH during heavily polluted events
in the winter, which is consistent with the main contribution of nitrate that we found in the BTH (Figure S4).

Although we filtered the features according to their relative importance and priori knowledge, collinearity still exists among the input features. Multicollinearity among features does not affect the performance of tree-based models like Random Forest and LightGBM (Belgiu and Drăguţ, 2016; Chen et al., 2016; Ke et al., 2017), but the contribution of a single feature might be slightly influenced by other features. Previous studies (Hou et al., 2022; Ye et al., 2022) have used ML to explain the
255 causes of air pollution and model bias, and although there was multicollinearity between the input features they used, they got reliable conclusions, showing the minimal impact of multicollinearity and the reliability of tree-based machine learning methods.

The main objective of this study was to diagnose the contributors to CMAQ simulation biases using machine learning, rather than for prediction. Since meteorology or emissions can only partially explain the simulation bias, a low $R^2$ is justified
when fitting the model with only meteorology or emissions variables (Figure 4). We designed a complementary experiment to measure the impact of the model itself by comparing popular regression models (including multiple linear regression, polynomial regression (degree:2), Random Forest, XGBoost, and LightGBM) with the same features ($PM_{2.5}$ components). All models show similar performance (Table A6), e.g., all models show lower $R^2$ in the winter in the BTH (0.16 - 0.4), and higher $R^2$ in the SCB region (0.7 - 0.8). This is side evidence that the low $R^2$ is more affected by the features than the model itself, as
the commonly used regression models can hardly obtain high $R^2$ with insufficient explanatory features (e.g., chemical component features in winter in BTH). Besides, LightGBM shows comparable accuracy to XGBoost but is faster and shows smaller accuracy gaps between training and testing with less overfitting.

Previous pollution prediction studies based on tree models usually added time-related features to describe the temporal pattern of pollutant changes to further improve the prediction ability, e.g., Wei et al. (2021a) improved the model performance
by adding temporal features of day of year and Unix timestamps. However, the inclusion of temporal features cannot provide any useful information about contributors of simulation biases instead it is difficult to attribute them to meteorological or emissions contributions.Therefore, temporal features were not included in our model. Besides, the ML bias diagnosis model constructed in this study is based entirely on local data and some temporal and regional processes influencing $PM_{2.5}$ concentrations are not considered in this study, such as vertical transport, long distance transport, which should be better
diagnosed in future work, and the main bias contributors of identified by variable importance are in good agreement with the current findings.

## 4 Conclusion

Based on artificial intelligence technology, this study systematically diagnoses the possible drivers of biases in $PM_{2.5}$ simulation from three perspectives of meteorology, chemical components, and emission sources. The relative importance of
280 multiple factors helps to understand the sources of simulation bias and the deficiencies of the CMAQ mechanisms. SOA is the main contributor to simulation biases among chemical components. $PM_{2.5}$ is more underestimated in dry weather. Among source sectors, residential contributed the most simulation bias for both PPM and SIA. These results provide valuable information for CMAQ model improvement from SOA and dust aerosol underestimation, meteorological field uncertainties, imperfection of the dry deposition scheme, and inaccurate residential emission inventories. As an efficient bias diagnosis
method, machine learning based methods provide valuable complements to traditional mechanism-based methods. This approach also greatly reduces the prior information for diagnosing simulation bias and efficiently identifies the important

contributors, so it can be easily extended to other CTMs models as well as other pollutants.

**Supporting**

Additional descriptions of the study domain, WRF-CMAQ simulation performance, concentrations and biases contribution of
$PM_{2.5}$ components and sectoral sources.

**Code/Data availability**

The data and code are publicly accessible in https://zenodo.org/record/7907626, including machine learning code for diagnosing CMAQ simulation bias and the corresponding training dataset. CMAQ is an open-source chemical transport model developed by the US Environmental Protection Agency, which can be downloaded at https://zenodo.org/record/1079898.

**Author contribution**

**Shuai Wang**: Methodology, Software, Writing - original draft. **Mengyuan Zhang**: Software, Validation. **Yueqi Gao**: Data curation, Visualization. **Peng Wang**: Methodology, Writing - reviewing and editing. **Qingyan Fu**: Writing - reviewing and editing. **Hongliang Zhang**: Conceptualization, Supervision, Writing - reviewing and editing.

**Competing interests**

The authors declare that they have no known competing financial interests or personal relationships that could have appeared to influence the work reported in this paper.

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

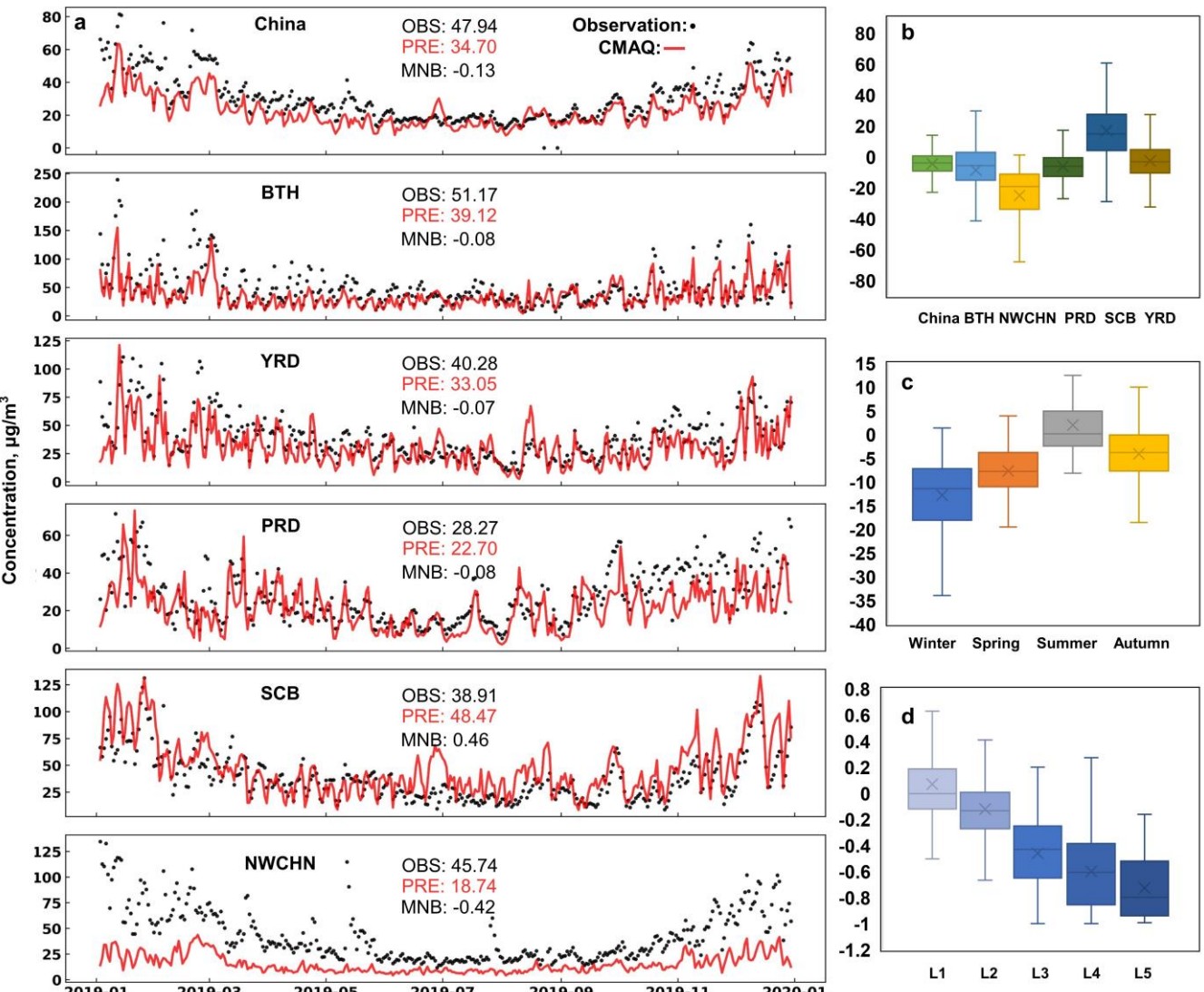

Figure 1. (a) The time series of observed (black) and CMAQ simulated (red) daily surface PM₂.₅ concentrations in China and five regions. Mean concentrations of the observed and simulated PM₂.₅ and MNB also shown in the inset. (b) Box plots of CMAQ simulated biases (simulated minus observed) for different regions. Crosses indicate average values and outliers are determined to be > 1.5 times of the upper quartile and < 1.5 times of the lower quartile. (c) Same as (b) but for four seasons. Spring, summer, autumn and winter are defined as March to May, June to August, September to November, December January and February, respectively. (d) Same as (b) but for different PM₂.₅ concentration levels (L1: [0, 35], L2: [35, 75], L3: [75, 115], L4: [115, 150], L5: [150, 1000]).

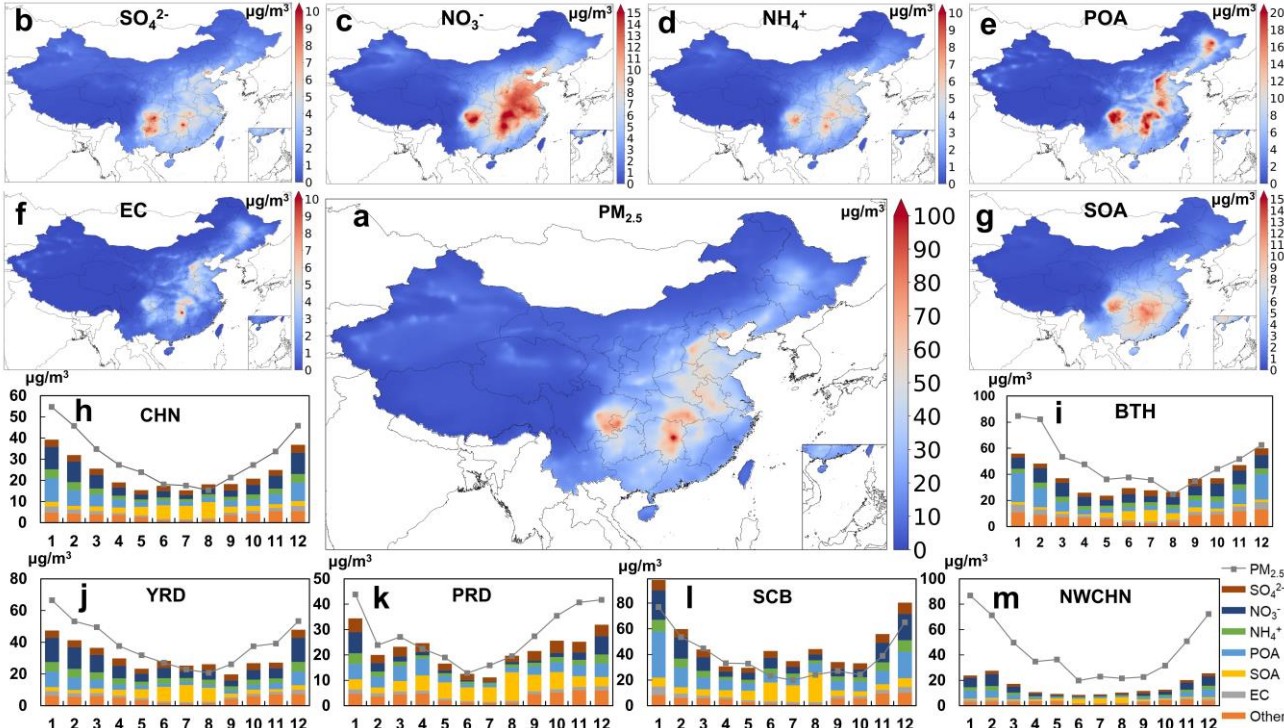

**Figure 2. Annual mean concentrations map (a - g) and monthly mean concentrations (h-m) of PM$_{2.5}$ and its components (SIA , POA, SOA, EC, and other components) for China and five key regions in 2019. Dotted lines in h-m indicate PM$_{2.5}$ observations**

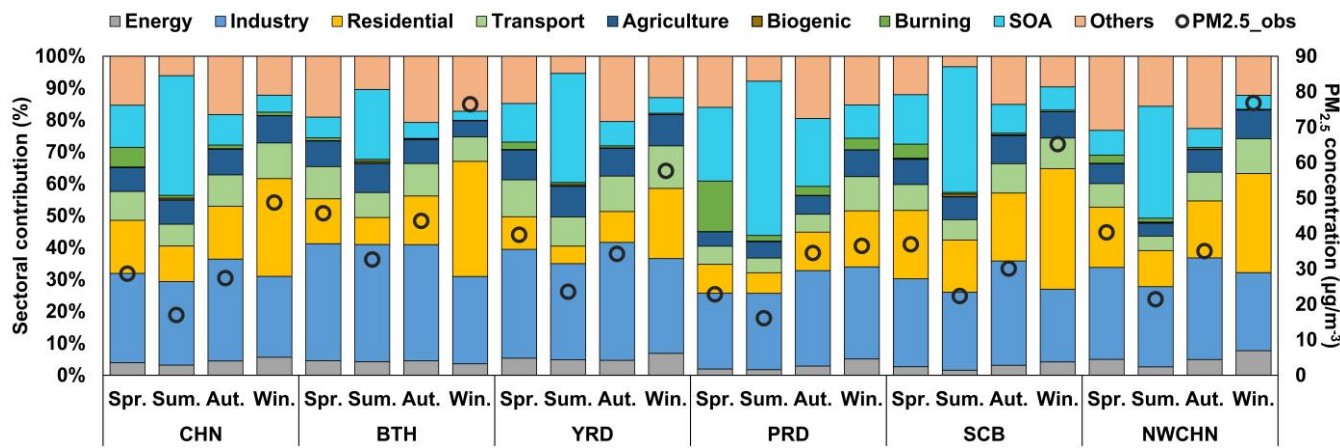

**Figure 3. Seasonal average fractional contributions from different sources to PM$_{2.5}$ concentrations (black circle on the right-hand axis) in China and five regions.**

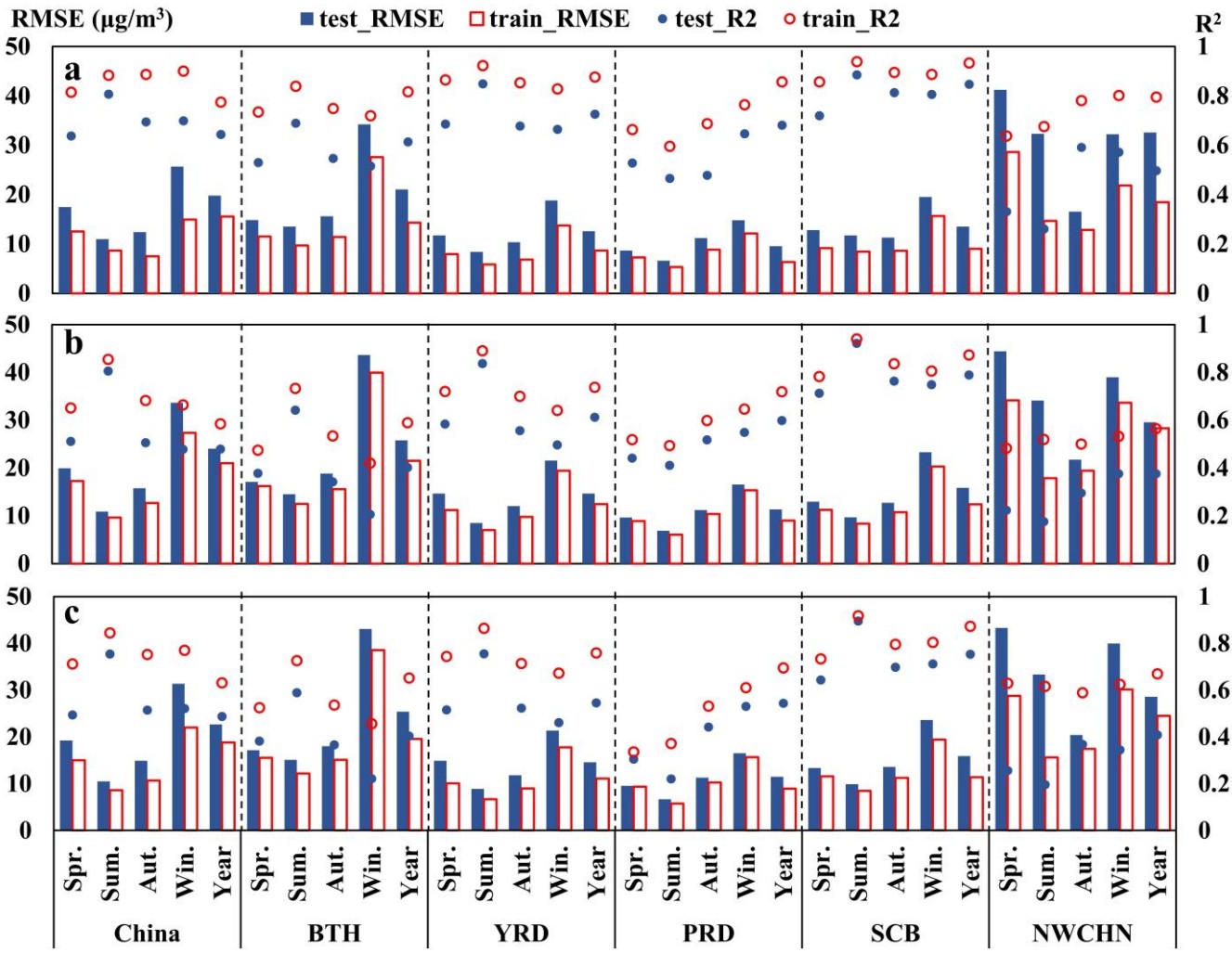

**Figure 4. Test results of CMAQ bias model training by meteorology (a), PM$_{2.5}$ components (b), and source sectors (c). RMSE unit: μg/m³.**

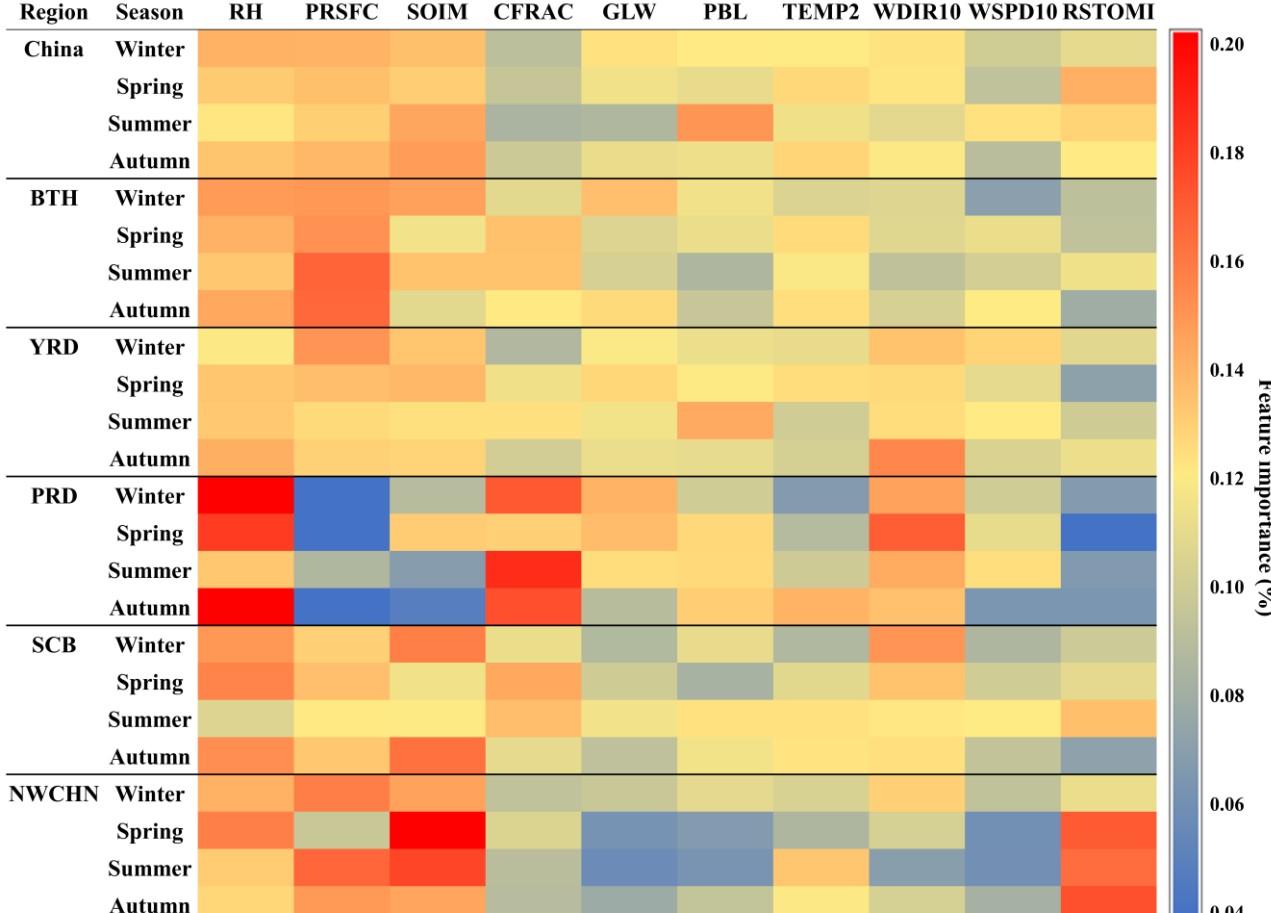

**Figure 5.** Contribution (%) of each meteorological factor to CMAQ simulation biases by region and season.

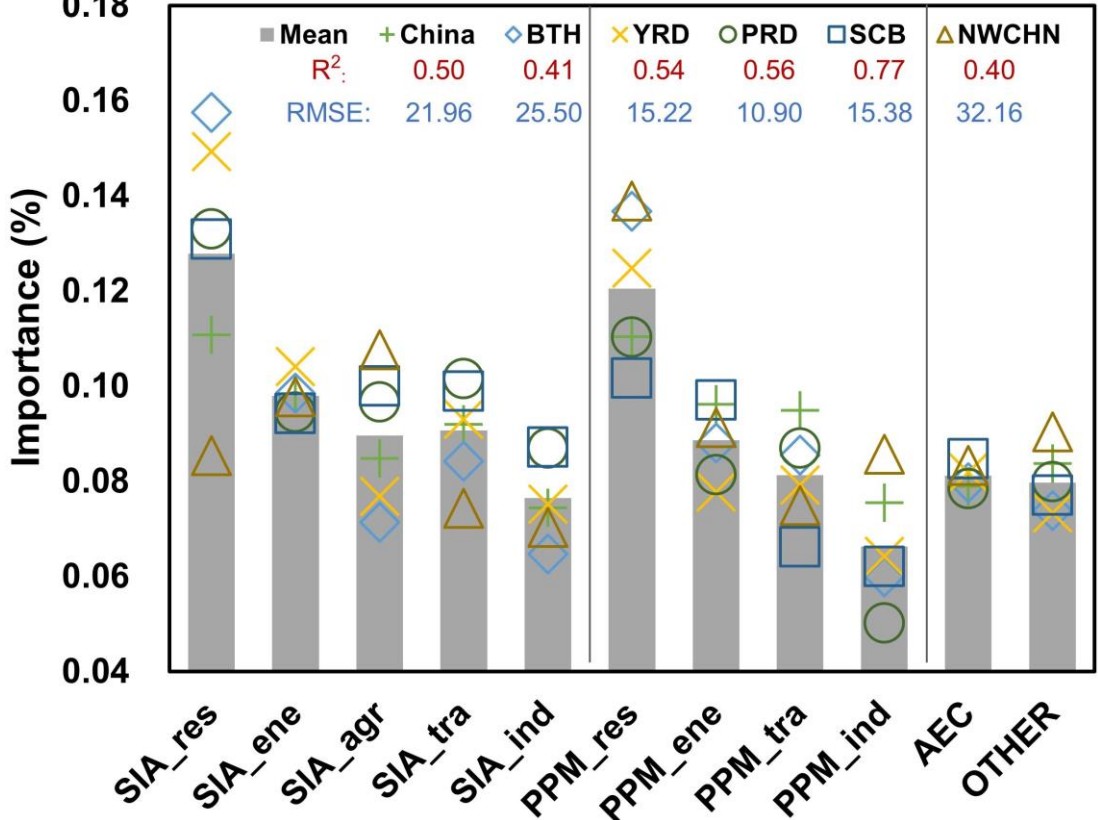

**Figure 6. Contribution (%) of each source sectors to CMAQ biases by region and season. res: residential, ene: energy, tra: transportation, agr: agriculture, ind: industry, AEC: elemental carbon, Other: other components.**