# Peer review of "Diagnosing drivers of $PM_{2.5}$ simulation biases in China from meteorology, chemical composition, and emission sources using an efficient machine learning method"

_EGUsphere, 2023_

## Author Response (AR1)

**Point-by-point responses to the comments & suggestions from the editor**

**Journal: Geoscientific Model Development**

**Manuscript ID: EGUSPHERE-2023-1531**

**Title: "Diagnosing drivers of PM2.5 simulation biases from meteorology, chemical composition, and emission sources using an efficient machine learning method"**

**Comments from the editor:**

I would like to point out an issue regarding the code availability in your manuscript and CMAQ. Currently, for CMAQ, you mention that it is available in a GitHub repository. GitHub repositories are not acceptable for scientific publication or long-term code archival, GitHub itself says it in its webpage, and we mention it in our Code and Data policy. Fortunately, for CMAQ, you have available Zenodo repositories too, such as https://zenodo.org/record/5213949. This way, please, look for the Zenodo repository corresponding to the CMAQ version that you use, post it replying to this comment, and in your manuscript cite it instead of the GitHub. If there is not a Zenodo repository for the version that you have used, you can upload the code yourself and create a new one.

- We appreciate the editor's comment on this manuscript. The CMAQ v5.0.2 code is publicly accessible at https://zenodo.org/record/1079898. We cited it in our manuscript.

**Changes in Lines 271-272:** "CMAQ is an open-source chemical transport model developed by the US Environmental Protection Agency, which can be downloaded at https://zenodo.org/record/1079898."

**Comments from Reviewer #1:**

This paper designed an efficient method based on machine learning for diagnosing the drivers of the Community Multiscale Air Quality (CMAQ) model biases in simulating PM2.5 concentrations from three perspectives of meteorology, chemical composition, and emission sources. The authors used source-oriented

CMAQ to diagnose the influences of different emission sources on PM2.5 biases. While the approach presented in the manuscript, particularly the emphasis on identifying biases in the CMAQ simulation of PM2.5 concentration, is innovative and distinct from conventional predictive models, a few issues should be addressed.

- We appreciate the comments from the reviewer, which help improve the manuscript. In the revision, we carefully revised the manuscript based on these comments.

**Major comments:**

1. Line 102: "Five-fold cross-validation method was used to evaluate the model fit and prediction ability (Browne, 2000)," and Line 143: "The ML models were trained separately for different regions and seasons, and a 5-fold cross-validation was used to measure the model performance (Figure 4)."

Cross-validation is employed primarily for model selection and hyperparameter tuning rather than evaluating the ML performance. To accurately evaluate the model's performance and generalization capabilities, it is essential to test it on a dataset it has never seen during training or validation. In addition, for clarification and completeness, the authors should provide a detailed explanation of why they chose 5-fold cross-validation and how they implemented this methodology in their study.

- Thanks for the comment. We carried out testing for the model used in this study. We randomly selected 20% of the data as the test set, and trained the model using a combination of meteorological, emission, and PM$_{2.5}$ components features, then predicted the simulated bias of the test set and compared it to the true bias (PM$_{2.5}$ from observations minus PM$_{2.5}$ from CMAQ simulations) (Figure S4). The model show an prediction R$^2$ of 0.68 and RMSE of 17.26 μg/m$^3$. We have added the corresponding results in Section 3.2. We added the reasons for choosing cross-validation and briefly introduced the cross-validation method that we used in Section 2.3.

**Changes in Lines 151-154:** "First, 20% of the data (not involved in training) were randomly selected for model evaluation (Figure S4). Training was performed

using a combination of PM$_{2.5}$ components, meteorological, and emission features. The model showed an prediction R$^2$ of 0.68 and RMSE of 17.26 μg/m$^3$."

**Changes in Lines 107-111:** "Cross-validation (CV) is an effective model validation method to prevent overfitting (Browne, 2000). To improve computational efficiency and enlarge the test dataset size, five-fold CV method was used to evaluate the model performance. The dataset was randomly divided into five parts, one was taken in turn as a test and the rest was used for training, which was repeated five times, and then the mean coefficient of determination (R$^2$) and the root mean square error (RMSE) were calculated."

2. Line 221: "In addition, the main objective of this study was diagnosing the contributors to CMAQ simulation biases using machine learning, therefore we did not pursue a very good model performance."

If the model's efficacy is insufficient, the interpretations and conclusions that can be drawn from it may be weakened. It is crucial to ensure that the model's predictions or interpretations are at least reasonably accurate. In addition, when using a specific model such as LightGBM, it would be beneficial to provide justification or evidence for why it outperforms other models such as Random Forest (Line 219) in this study. Such a justification can lend more credibility to the findings and insights derived from the model.

- Thanks for the comment. We use machine learning to capture the relationship between simulation bias and input variables, rather than for prediction. Since meteorology or emissions can only partially explain the simulation bias, a poor R$^2$ is justified when fitting the model with only meteorology or emissions variables. R$^2$ here indicates how well the input variables explain the results, and a low R$^2$ indicates a minor influence of the input variables to the simulation bias. We added the corresponding description in Section 3.3.

The LightGBM model is an optimized Gradient Boosting Decision Tree (GBDT). Compared to XGBoost, a widely used GBDT, LightGBM uses Histogram's decision tree algorithm along with Gradient-based One-Side Sampling (GOSS), which

saves memory and computation time (Ke et al., 2017). Three tree-based models, Random Forest, XGBoost, and LightGBM, were compared in our previous study (Wang et al., 2023). We found that using the same input data and hyperparameters, LightGBM is as accurate as XGBoost, but faster and less overfitting (the difference in accuracy between training and testing), so here we chose the LightGBM model for simulation bias diagnosing. We added the reasons for choosing LightGBM in Section 2.3.

**Changes in Lines 242-246:** "In addition, the main objective of this study was diagnosing the contributors to CMAQ simulation biases using machine learning, rather than for prediction. Since meteorology or emissions can only partially explain the simulation bias, a low $R^2$ is justified when fitting the model with only meteorology or emissions variables (Figure 4), which indicates a minor influence of the input variables to the simulation bias."

**Changes in Lines 93-99:** "The LightGBM model is an optimized Gradient Boosting Decision Tree (GBDT) (Ke et al., 2017), and has shown accurate performance in many fields (Wei et al., 2021; Yan et al., 2021; Sun et al., 2020; Liang et al., 2020). Compared to XGBoost, a widely used GBDT, LightGBM uses Histogram's decision tree algorithm along with Gradient-based One-Side Sampling (GOSS), which saves memory and computation time (Ke et al., 2017). Three tree-based models, Random Forest, XGBoost, and LightGBM, were compared in our previous study (Wang et al., 2023). Using the same input data and hyperparameters, LightGBM is as accurate as XGBoost, but faster and less overfitting (the difference in accuracy between training and testing), so the lightGBM model was used to diagnose $PM_{2.5}$ simulation biases in this study."

**Minor Comments:**

1. The study area for this manuscript is China; therefore, "in China" should be added to the title.

- Thanks for the comment. We have changed the title to specify the study area of

China.

**Changes in the title:** "Diagnosing drivers of $PM_{2.5}$ simulation biases in China from meteorology, chemical composition, and emission sources using an efficient machine learning method"

2. Line 56: Chemical components constitute PM2.5, so they would be considered as labels. Why are they used as features? Additionally, wouldn't the linear summation of all these chemical components essentially represent PM2.5? If I have misunderstood any aspect, it would be helpful if the author could explain it.

- Thanks for the comment. Indeed, chemical components constitute $PM_{2.5}$. So, the simulation bias of total $PM_{2.5}$ should be attributed to the specific components. Using the components as input features to fit the total simulation bias can tell us which components have a large simulation bias. We have added a corresponding note in the manuscript.

**Changes in Lines 157-158:** "Using $PM_{2.5}$ components as input features to fit the total simulation bias can tell us which components have a large simulation bias."

3. Line 62: How are "problematic data points" defined?

- Thanks for the comment. We are sorry for the unclear presentation. "problematic data points" here mean extreme value, records of $PM_{2.5}$ exceeds $PM_{10}$, and days with less than 20-hour records. We have added this description in Section 2.1.

**Changes in Lines 62-63:** "The daily observations data <0.1 % quantile and >99.9 % quantile, $PM_{2.5}$ exceeds $PM_{10}$, and days with less than 20 valid hourly records are excluded."

4. Line 78: Table S1 is "Summary of the WRF model variables used in this study." The list of PM2.5 components simulated by CMAQ is not available in Table S1.

- Thanks for the comment. We are sorry for our carelessness. The list of $PM_{2.5}$

components simulated by CMAQ has been added to Table A1.

5. Line 96: Could you please clarify what is meant by "three combinations of input variables"? Does this refer to pairwise combinations of the categories (e.g., "meteorological factors" + "chemical components") or something else?

- Thanks for the comment. Three combinations of input variables mean meteorological factors, chemical components, and emission sources, that is, we trained the ML model for three times with three categories of variables separately. By doing so, the sources of simulation bias are analyzed from three perspectives: meteorological, emission, and components. We have added this description in Section 2.3.

**Changes in Lines 101-102:** "Three categories of input variables were designed to separately fit the simulation biases: meteorological factors, chemical components, and emission sources."

6. Line 107: It should be "Observed PM2.5 concentrations".

- Thanks for the comment. We have modified the corresponding descriptions and examined the manuscript carefully.

**Changes in Lines 116:** "Observed $PM_{2.5}$ concentrations were higher in BTH (51.172 μg/m$^3$) and lower in PRD (28.273 μg/m$^3$)."

7. Figure 3: The left axis features a stacked bar plot for sectoral contribution (with a maximal y-value of 100%), whereas the right axis represents PM2.5 concentration using a scatter plot. However, the areas in which the scatter points overlap with the bars do not provide clear information, making the use of dual axes potentially misleading.

- Thanks for the comment. We have modified Figure 3 and Figure S3, and replaced the solid circles with hollow circles in the PM scatter plot to make the figure clearer.

8. Figure 4: Some models have a training R2 lower than 0.6. This suggests that these models might be underfitting (please see my "major comments").

- Thanks for the comment. We use machine learning to capture the relationship between simulation bias and input variables, rather than for prediction. Since meteorology or emissions can only partially explain the simulation bias, a poor $R^2$ is justified when fitting the model with only meteorology or emissions variables. $R^2$ here indicates how well the input variables explain the results, and a low $R^2$ in some regions and seasons indicates a minor influence of the input variables to the simulation bias. Also, part of the reason could be attributed to poor model fit due to data quantity and quality. We added the corresponding description in Section 3.3.

**Changes in Lines 242-246:** "In addition, the main objective of this study was diagnosing the contributors to CMAQ simulation biases using machine learning, rather than for prediction. Since meteorology or emissions can only partially explain the simulation bias, a low $R^2$ is justified when fitting the model with only meteorology or emissions variables (Figure 4), which indicates a minor influence of the input variables to the simulation bias."

**Comments from Reviewer #2:**

The article describes an interesting method to determine the origin of the bias of a CTM using a ML algorithm. It applies this method to an interesting case and allows to determine what sector biases come from. This method has the potential to be applied in similar studies. An extensive bibliography is provided, enabling the reader to find more details when necessary. It would be nice to have a comparison of the results of this method, in the case studied, with those of other methods. A short reminder of the basics of the algorithms used (even if the references provided do that in details) would have been welcome.

- We appreciate the comments from the reviewer, which help us a lot. We compare studies of CTM simulation bias identification in China using different methods. We obtained many consistent conclusions, e.g., systematic underestimation of SOA, significant contribution of primary $PM_{2.5}$ emissions, and inaccurate

simulation of nitrate in winter in Beijing. We added a discussion in Section 3.3.

The LightGBM model is an optimized Gradient Boosting Decision Tree (GBDT). Compared to XGBoost, a widely used GBDT, LightGBM uses Histogram's decision tree algorithm along with Gradient-based One-Side Sampling (GOSS), which saves memory and computation time (Ke et al., 2017). We added the introduction of LightGBM in Section 2.3.

**Changes in Lines 230-237:** "Huang et al. (2019) used a new reduced-form model coupled with a higher-order decoupled direct method and stochastic response surface model to identify sources of uncertainty in CMAQ simulations. An analysis for the PRD of China in spring 2013 revealed a systematic underestimation of SOA and identified wind speed and primary $PM_{2.5}$ emissions as key sources of uncertainties in $PM_{2.5}$ simulations, which is consistent with the results identified using LightGBM in this study. Aleksankina et al. (2019) identified $PM_{2.5}$ simulation bias in Europe using optimised Latin hypercube sampling and also demonstrated the important impact of primary emissions on $PM_{2.5}$ simulation uncertainties. Liu and Xing (2022) used a fully connected neural network to identify $PM_{2.5}$ simulations biases and found that $NO_2$ is the main contributor in BTH during heavy polluted events in the winter, which is consistent with the main contribution of nitrate that we found in the BTH (Figure S5)."

**Changes in Lines 93-96:** "The LightGBM model is an optimized Gradient Boosting Decision Tree (GBDT) (Ke et al., 2017), and has shown accurate performance in many fields (Wei et al., 2021; Yan et al., 2021; Sun et al., 2020; Liang et al., 2020). Compared to XGBoost, a widely used GBDT, LightGBM uses Histogram's decision tree algorithm along with Gradient-based One-Side Sampling (GOSS), which saves memory and computation time (Ke et al., 2017)."

**Specific comments**

L17: "model biases", bias when the model is compared to observations ? It could be precised.

- Thanks for the comment. We have modified the corresponding descriptions and

examined the manuscript carefully.

**Changes in Lines 16-19:** "In this study, an efficient method based on machine learning (ML) was designed to diagnose the drivers of the Community Multiscale Air Quality (CMAQ) model biases compared to observations in simulating PM$_{2.5}$ concentrations from three perspectives of meteorology, chemical composition, and emission sources."

L20: even if we are in the abstract, the expressions "fitting ability" should be precised.

- Thanks for the comment. We have modified the expressions and explained it with plain language.

**Changes in Lines 19-21:** "The ML model can capture the complex relationship between input variables and simulation bias well with a small performance gap between training and validation."

L29: could you remind the reader the definition of the acronyms PPM and SIA/SOA ?

- Thanks for the comment. We have clarified the definition of PPM and SIA/SOA.

**Changes in Lines 30-31:** "Fine particulate matter (PM$_{2.5}$) is a complex mixture of primary particulate matters (PPM) and secondary inorganic/organic components (SIA/SOA), with adverse effects on public health and ecosystems."

L42: which ML methods ?

- Thanks for the comment. We have added the popular ML methods used before: Random Forest and XGBoost.

**Changes in Lines 44-46:** "Recently machine learning (ML) methods, like Random Forest and XGBoost, have been widely used in environmental science researches due to their simple structure, fast speed and ability to deal with no-linear relationships"

L41-52 Could you quantifiy the gain in computing resources you achieve using your

ML method to the methods using Monte Carlo or Latin hypercube sampling ? Have you tried these other methods and compared their results with the method described in this article ?

- Thanks for the comment. The LightGBM model used in this study is very fast, taking only a few tens of seconds for a single core to train at a time, and with low memory usage (depending on the size of the training dataset), the size of this training dataset is around 360,000 rows, 57 columns, in 64-bit floating-point format, and only requires about 160 MB of memory, making it ready to run on a laptop computer. However, Monte Carlo-based methods require multiple runs of the chemical transport model for sensitivity testing, which can more accurately identify factors that cause bias, but are computationally demanding and typically cannot be run on personal computers, relying on high-performance multi-core computers.

We compare studies of CTM simulation bias identification in China using different methods. We obtained many consistent conclusions, e.g., systematic underestimation of SOA, significant contribution of primary $PM_{2.5}$ emissions, and inaccurate simulation of nitrate in winter in Beijing. We added a discussion in Section 3.3.

**Changes in Lines 230-237:** "Huang et al. (2019) used a new reduced-form model coupled with a higher-order decoupled direct method and stochastic response surface model to identify sources of uncertainty in CMAQ simulations. An analysis for the PRD of China in spring 2013 revealed a systematic underestimation of SOA and identified wind speed and primary $PM_{2.5}$ emissions as key sources of uncertainties in $PM_{2.5}$ simulations, which is consistent with the results identified using LightGBM in this study. Aleksankina et al. (2019) identified $PM_{2.5}$ simulation bias in Europe using optimised Latin hypercube sampling and also demonstrated the important impact of primary emissions on $PM_{2.5}$ simulation uncertainties. Liu and Xing (2022) used a fully connected neural network to identify $PM_{2.5}$ simulations biases and found that $NO_2$ is the main contributor in BTH during heavy polluted events in the winter, which is consistent with the main contribution of nitrate that we found in the BTH (Figure S5)."

L63: could you precise what you mean by "same simulation grid" ? Could this grid be defined.

- Thanks for the comment. CMAQ simulation was conducted with a 36 km horizontal resolution. For areas with a high density of observation sites, such as Beijing, several sites may be located in the same 36km*36km grid, in which case the average of several sites in the same grid will be calculated. We have clarified it.

**Changes in Lines 63-65:** "For observation sites located on the same CMAQ simulation grid (36 km × 36 km), average $PM_{2.5}$ concentrations of these sites were calculated to compare with CMAQ simulation."

L80 would it be possible to me give more detail on the source apportionment method ?

- Thanks for the comment. PPM from different source sectors are tracked by non-reactive tracers ($10^{-5}$ of the PPM emission rates). The concentrations of PPM from given sources are then calculated by multiplying the tracer with $10^5$. The contributions of source sectors to SIA are quantified using specific reactive tagged tracers. Specifically, $NO_x$, $SO_2$, and $NH_3$ from different sources were tracked separately through a series of chemical and physical processes involved in SIA formation. We added the corresponding description in **Section 2.2**.

**Changes in Lines 83-86:** "PPM from different source sectors are tracked by non-reactive tracers ($10^{-5}$ of the PPM emission rates). The concentrations of PPM from given sources are then calculated by multiplying the tracer with $10^5$. The contributions of source sectors to SIA are quantified using specific reactive tagged tracers. Specifically, $NO_x$, $SO_2$, and $NH_3$ from different sources were tracked separately through a series of chemical and physical processes involving in SIA formation."

L90-102: a brief presentation of the algorithms would be interesting (see general comments).

\- Thanks for the comment. The LightGBM model is an optimized Gradient Boosting Decision Tree (GBDT) (Ke et al., 2017). Compared to XGBoost, a widely used GBDT, LightGBM uses Histogram's decision tree algorithm along with Gradient-based One-Side Sampling (GOSS), which saves memory and computation time. We added the description of LightGBM in Section 2.3.

**Changes in Lines 93-96:** "The LightGBM model is an optimized Gradient Boosting Decision Tree (GBDT) (Ke et al., 2017), and has shown accurate performance in many fields (Wei et al., 2021; Yan et al., 2021; Sun et al., 2020; Liang et al., 2020). Compared to XGBoost, a widely used GBDT, LightGBM uses Histogram's decision tree algorithm along with Gradient-based One-Side Sampling (GOSS), which saves memory and computation time (Ke et al., 2017)."

L90 the only citation for this method in the bibliography is from a conference. Why not mentioning

  "Ke, G., Meng, Q., Finley, T., Wang, T., Chen, W., Ma, W., … Liu, T.-Y. (2017). Lightgbm: A highly efficient gradient boosting decision tree. Advances in Neural Information Processing Systems, 30, 3146–3154." ?

\- Thanks for the comment. We learned about lightGBM method from the conference paper, so we cited it. We have changed to a more formal citation provided by the reviewer.

**Changes in Lines 93:** "The LightGBM model is an optimized Gradient Boosting Decision Tree (GBDT) (Ke et al., 2017)"

L102 even if it defined in the reference, it would be nice to remind (briefly) what the X-fold cross-validation method consists in.

\- Thanks for the comment. We have briefly introduced the cross-validation method that we used in Section 2.3.

**Changes in Lines 107-111:** "Cross-validation (CV) is an effective model validation method to prevent overfitting (Browne, 2000). To improve computational efficiency and enlarge the test dataset size, five-fold CV method was used to evaluate

the model performance. The dataset was randomly divided into five parts, one was taken in turn as a test and the rest was used for training, which was repeated five times, and then the mean coefficient of determination ($R^2$) and the root mean square error (RMSE) were calculated."

L135-136: why just analyzing source sectors of SIA and not SOA ?

    - Thanks for the comment. The formation mechanism of SOA is complicated and currently incomplete, and the emission of precursor VOCs has high uncertainty, therefore, we did not track sources of SOA. We have added a corresponding description in Section 2.2.

    **Changes in Lines 87-88:** "The source of SOA was not traced due to the complex and currently imperfect mechanism of SOA formation and the high uncertainty in the precursor VOCs emissions (Liu et al., 2021; Hu et al., 2017)."

L156: What is imperfect: the pathways or their current knowledge ?

    - Thanks for the comment. We apologize for the lack of clarity, but "imperfect" here refers to the imperfect nitrate mechanism (e.g. non-homogeneous oxidation) in the SAPRC11 mechanism that we used. We have clarified this point

    **Changes in Lines 167-170:** "Nitrate contribution to simulation bias further implies the inaccuracy of nitrate simulations, which can relate to the imperfect pathways of nitrate production (e.g., non-homogeneous oxidation) in SAPRC11 (that we used) and the uncertainties of nitrate precursor emission inventories in winter (Xu et al., 2019; Zhang et al., 2018; Carter and Heo, 2013)."

L196: could you be more specific about the subsurface conditions ?

    - Thanks for the comment. Here "subsurface conditions" mean the land surface properties, the rate of dry deposition is closely related to land cover type. We have added a corresponding note in Section 3.2.

    **Changes in Lines 207-210:** "Dry deposition is a critical but highly uncertain sink for aerosols, which depends on the chemical and physical properties of aerosols,

and be influenced by land surface properties and meteorological conditions (Shu et al., 2022). Different land-use types (e.g., vegetation, deserts, and snow) have significantly different abilities to capture particulate matter."

L206: I don't see the values mentioned for R2 and RMSE (0.53 and 20.18) in figure 6. What do they relate to ?

- Thanks for the comment. The values mentioned for $R^2$ and RMSE are from Figure 4, and we modified the expression.

**Changes in Lines 217-218:** "In China and five key regions, sectoral sources were able to fit the simulation bias well, with mean $R^2$ and RMSE of 0.53 % and 20.18 µg/m$^3$ (Figure 4)."

**Technical corrections**

L25: "contribution" => "contribution to this bias".

- Thanks for the comment. We have modified the expression accordingly, and checked the manuscript.

**Changes in Lines 25-26:** "Both primary and secondary inorganic components from residential sources showed the largest contribution to this bias (12.05 % and 12.78 %), implying large uncertainties in this sector."

L64: I would suggest to change the formulation: "Analysis focused on nationwide as well as several interested regions" => "Analysis has been carried out on several regions of interest and on all China."

- Thanks for the comment. We have modified the expression accordingly to make it clearer.

**Changes in Lines 65-66:** "Analysis has been carried out on several haze-polluted regions and on all China (Figure S1)"

L65: could you justify the choice of the regions ?

- Thanks for the comment. We selected sub-regions according to the severity of

haze pollution. We have modified the expression "interest" to "haze-polluted".

**Changes in Lines 65-66:** "Analysis has been carried out on several haze-polluted regions and on all China (Figure S1)"

L79: "conducted over" => "carried out"

- Thanks for the comment. We have modified the expression accordingly to make it clearer.

**Changes in Lines 79-80:** "The CMAQ simulation (36 km×36 km) was carried out in mainland China and surrounding regions in 2019."

L107    "higher" => "highest", "lower" => "lowest"

- - Thanks for the comment. We have modified the expression accordingly and double-checked the manuscript to make sure it is correct.

**Changes in Lines 116:** "Observed $PM_{2.5}$ concentrations were highest in BTH (51.172 μg/m$^3$) and lowest in PRD (28.273 μg/m$^3$)."

L173: "the stationary" => "a stationary"

- Thanks for the comment. We have modified the expression accordingly to make it clearer.

**Changes in Lines 183-184:** "High pressure systems are connected to a stationary weather, which is unfavorable for $PM_{2.5}$ dispersion."

L175: "the uncertain" => "the uncertainties"

- Thanks for the comment. We are sorry for our carelessness and have modified the expression accordingly and double-checked the manuscript to make sure it is correct.

**Changes in Lines 187-188:** "Contribution of wind direction in YRD may also related to the uncertainties of WRF simulation."

L177: "Earth's radiation receipts" : I would prefer    "radiation received by the Earth"

- Thanks for the comment and suggestion. We have modified the expression accordingly

**Changes in Lines 189-190:** "In addition to directly changing the radiation received by the earth through scattering and absorbing"

L181: "shown the dominant" => "showed the dominant".

- Thanks for the comment. We are sorry for our carelessness and have modified the expression accordingly. We double-checked the grammar of the manuscript to make sure it is correct.

**Changes in Lines 193-194:** "Previous study showed the dominant role of cloud chemistry in aerosol-cloud interactions in southern China"

L183: "the missing" => "the lack"

- Thanks for the comment and suggestion. We have modified the expression accordingly

**Changes in Lines 194-195:** "Therefore, the influence of cloud cover on simulation biases in YRD can attributed to the lack of aerosol feedback mechanism."

L185: "can associate" => "can be associated"

- Thanks for the comment. We are sorry for our carelessness and have modified the expression accordingly. We double-checked the grammar of the manuscript to make sure it is correct.

**Changes in Lines 197:** "These factors can be associated with ground-level sand rise and dust emission."

L186: "attributed" => "be attributed"

- Thanks for the comment. We are sorry for our carelessness and have modified the expression accordingly. We double-checked the grammar of the manuscript to make sure it is correct.

**Changes in Lines 197:** "Underestimation of dust aerosol in NWCHN can be attributed to emission"

L196: "influenced" => "is influenced"

    - Thanks for the comment. We are sorry for our carelessness and have modified the expression accordingly. We double-checked the grammar of the manuscript to make sure it is correct.

    **Changes in Lines 207-208:** "Dry deposition is a critical but highly uncertain sink for aerosols, which depends on the chemical and physical properties of aerosols, and be influenced by subsurface and meteorological conditions"

L199 " the underestimates" => "an underestimation"

    - Thanks for the comment. We have modified the expression accordingly

    **Changes in Lines 211-212:** "Recent studies for the United States also showed an underestimation for $PM_{10}$ concentrations."

Figure S4 : difficult to distinguish between the different shades of red/pink. The use of another color scale (with differerent colors) would be clearer.

    - Thanks for the comment. We have modified the Figure S4 (renumbered as Figure S5) with blue-yellow-red color bar to make the figure clearer.

Figure 4: Bottom of figure 4: RSME => RMSE

    - Thanks for the comment. We are sorry for our carelessness and have modified the expression in Figure 4 accordingly.

Figure 5 : difficult to distinguish between the different shades of red/pink. The use of another color scale (with differerent colors) would be clearer.

    - Thanks for the comment. We have modified the Figure 5 with blue-yellow-red color bar to make the figure clearer.

Problem in numbering: we have the same notation for figures (S1,...) and tables (S1,...) It would be better to have a different notation for the tables and for the figures

- Thanks for the comment. We have used different notation for figures (S1,…) and tables (A1,…), and modified the corresponding references in the manuscript.

Reference

Aleksankina, K., Reis, S., Vieno, M., and Heal, M. R.: Advanced methods for uncertainty assessment and global sensitivity analysis of an Eulerian atmospheric chemistry transport model, Atmos. Chem. Phys., 19, 2881-2898, 2019.

Browne, M. W.: Cross-validation methods, Journal of mathematical psychology, 44, 108-132, 2000.

Carter, W. P. and Heo, G.: Development of revised SAPRC aromatics mechanisms, Atmos. Environ., 77, 404-414, 2013.

Hu, J., Wang, P., Ying, Q., Zhang, H., Chen, J., Ge, X., Li, X., Jiang, J., Wang, S., and Zhang, J.: Modeling biogenic and anthropogenic secondary organic aerosol in China, Atmos. Chem. Phys., 17, 77-92, 2017.

Huang, Z., Zheng, J., Ou, J., Zhong, Z., Wu, Y., and Shao, M.: A Feasible Methodological Framework for Uncertainty Analysis and Diagnosis of Atmospheric Chemical Transport Models, Environ. Sci. Technol., 53, 3110-3118, 10.1021/acs.est.8b06326, 2019.

Ke, G., Meng, Q., Finley, T., Wang, T., Chen, W., Ma, W., Ye, Q., and Liu, T.-Y.: Lightgbm: A highly efficient gradient boosting decision tree, Advances in neural information processing systems, 30, 2017.

Liang, W., Luo, S., Zhao, G., and Wu, H.: Predicting hard rock pillar stability using GBDT, XGBoost, and LightGBM algorithms, Mathematics, 8, 765, 2020.

Liu, J. and Xing, J.: Identifying Contributors to PM2.5 Simulation Biases of Chemical Transport Model Using Fully Connected Neural Networks, Journal of Advances in Modeling Earth Systems, 15, 2022.

Liu, J., Chu, B., Chen, T., Zhong, C., Liu, C., Ma, Q., Ma, J., Zhang, P., and He, H.: Secondary organic aerosol formation potential from ambient air in Beijing: effects of atmospheric oxidation capacity at different pollution levels, Environ. Sci. Technol., 55, 4565-4572, 2021.

Shu, Q., Murphy, B., Schwede, D., Henderson, B. H., Pye, H. O. T., Appel, K. W., Khan, T. R., and Perlinger, J. A.: Improving the particle dry deposition scheme in the CMAQ photochemical modeling system, Atmos. Environ., 289, 119343, https://doi.org/10.1016/j.atmosenv.2022.119343, 2022.

Sun, X., Liu, M., and Sima, Z.: A novel cryptocurrency price trend forecasting model based on LightGBM, Finance Research Letters, 32, 101084, 2020.

Wang, S., Wang, P., Zhang, R., Meng, X., Kan, H., and Zhang, H.: Estimating particulate matter concentrations and meteorological contributions in China during 2000–2020, Chemosphere, 330, 138742, https://doi.org/10.1016/j.chemosphere.2023.138742, 2023.

Wei, J., Li, Z., Pinker, R. T., Wang, J., Sun, L., Xue, W., Li, R., and Cribb, M.: Himawari-8-derived diurnal variations in ground-level PM2.5 pollution across China using the fast space-time Light Gradient Boosting Machine (LightGBM), Atmos. Chem. Phys., 21, 7863-7880, 10.5194/acp-21-7863-2021, 2021.

Xu, Q., Wang, S., Jiang, J., Bhattarai, N., Li, X., Chang, X., Qiu, X., Zheng, M., Hua, Y., and Hao, J.: Nitrate dominates the chemical composition of PM2. 5 during haze event in Beijing, China, Science of the Total Environment, 689, 1293-1303, 2019.

Yan, J., Xu, Y., Cheng, Q., Jiang, S., Wang, Q., Xiao, Y., Ma, C., Yan, J., and Wang, X.: LightGBM: accelerated genomically designed crop breeding through ensemble learning, Genome Biology, 22, 1-24, 2021.

Zhang, R., Sun, X. S., Shi, A. J., Huang, Y. H., Yan, J., Nie, T., Yan, X., and Li, X.: Secondary inorganic aerosols formation during haze episodes at an urban site in Beijing, China, Atmos. Environ., 177, 275-

282, 10.1016/j.atmosenv.2017.12.031, 2018.

---

## Referee Report (RR1)

Review for manuscript: *Diagnosing drivers of PM2.5 simulation biases in China from meteorology, chemical composition, and emission sources using an efficient machine learning method*

Summary: In this manuscript, the author introduces LightGBM, a tree-based regression method, as a powerful tool for evaluating the performance of the Community Multiscale Air Quality (CMAQ) model. The primary focus is on diagnosing the CMAQ's effectiveness in pinpointing the predominant contributing factors responsible for prediction bias, particularly in relation to the prediction of PM 2.5 concentration. To comprehensively assess potential biases associated with each source, LightGBM is employed to conduct separate time series regressions for features grouped into three major sources.

Major comments:

After reading the manuscript, I think some major comments from Anonymous Reviewer #1 of last round are still not adequately addressed. In my opinion, the author should clearly address the following aspects:

1. Dataset setting:
   a. Provide a clear description of the 350,000 valid observations, specifying whether it represents the sum of all time series data points across multiple monitoring stations. Clearly state the methodology for training and testing data separation.
   b. Clarify how random samples of observations are selected. Specify whether the 20% random sampling is performed at the station level or across all stations in the region of interest.
   c. Time series data usually cannot be directly learned through tree-based model without additional pre-processing/feature engineering. Discuss the absence of data preparation and feature engineering before feeding data into tree-based models. If temporal structure is considered negligible, provide justification; otherwise, explain the approach taken to handle temporal aspects.
2. Tree-based model justification
   a. In the section (L95-100), provide examples of similar applications in terms of dataset, model, and research area. Demonstrate why tree-based methods are suitable for the specific dataset. Justify the selection of tree-based models beyond considerations of memory and computation time.
   b. Introduce a discussion on multicollinearity in the methodology section.
3. Cross validation
   a. Clearly explain how cross-validation is performed and provide a statement on how the two metrics ($R^2$ and RMSE) influence model selection decisions. Clearly articulate the criteria for jointly considering model performance using these two metrics.

Minor comments:
L15. Clarify the term "efficient" to provide a precise understanding within the context of this study.

L16. Instead of broadly referring to "machine learning," explicitly specify that LightGBM is a tree-based method. Additionally, consider breaking the sentence into two for enhanced readability.

L20. Reevaluate the assertion that an R^2 value of 0.68 constitutes good performance. Provide references from existing literature to substantiate this claim. Additionally, the relative performance gap of 0.16 is about 23.5% of 0.68, which might not be compelling enough; its significance in the context of overfitting and the ability to be applied to other fields is weak.

L65. Revise the description of "valid" observations to explicitly convey that these observations adhere to the quality control criteria outlined in L62-64. Reorganize the sentences for better coherence.

L76. Consider either elaborating on the model's enhancements or removing the sentence for conciseness.

L80-82. Clearly indicate that CMAQ is employed for simulating PM 2.5 components when introducing the model. Adjust the sequence of information to improve logical flow.

L86. Enhance the fluency by adding a connecting word at the beginning of the sentence.

L119-120. Define "success" in quantitative or qualitative terms to provide a clearer understanding of the criteria for evaluating success.

L170. Remove the extra period before the citation.

L245. Replace "Features collinearity" with "Multicollinearity among features."

Table A6. Use bold font to highlight the best metric performance. Additionally, if XGB and LGB exhibit similar performance, with XGB slightly superior, consider including computational time as an additional metric to justify the preference for LGB over XGB.

---

## Author Response (AR2)

**Point-by-point responses to the comments & suggestions from the editor**

**Journal: Geoscientific Model Development**

**Manuscript ID: EGUSPHERE-2023-1531**

**Title: "Diagnosing drivers of PM₂.₅ simulation biases in China from meteorology, chemical composition, and emission sources using an efficient machine learning method"**

**Comments from the editor:**

Please read the reviewer's comments and reply accordingly.

- We appreciate the editor's efforts on this manuscript. In the revision, we carefully revised the manuscript based on reviewers' comments.

**Comments from Reviewer #1:**

The authors did not adequately address my comments.

- Thank you for your feedback and for taking the time to review our manuscript. We appreciate your input, and we apologize if it seemed that we did not adequately address your comments in our previous revision. We take your comments seriously and are committed to improving our manuscript to meet your expectations. we have made the necessary revisions and provided a detailed response to each of your comments in this revision. We are dedicated to producing a high-quality manuscript, and your feedback is invaluable in achieving that goal. Thank you again for your time and consideration. We look forward to working closely with you to address your concerns and make the necessary improvements.

1. A separate test dataset, which is excluded from the cross-validation process, is typically employed to validate the mode's performance. However, the test data were no provided by the authors. Ref: https://en.wikipediaorg/wiki/Training,validation,and test data sets#Cross-validation

- Thanks for the comment. We randomly divided a test set (20% of total data) that was not involved in the training and CV hyperparameter selection process, and a separate test dataset has been updated to zendo (https://zenodo.org/records/10283739). Hyperparameter selection and further model training were performed using the training dataset. The hyperparameters selected using only the training data are consistent with the hyperparameters we previously selected using all the data. The model was tested using a combination of meteorological, emission, and $PM_{2.5}$ components features in the test set (Figure S4). The model shows a prediction $R^2$ of 0.68 and RMSE of 17.26 µg/m$^3$. We have added the corresponding results in Section 3.2.

**Changes in Lines 155-158:** "First, 20% of the data (not involved in training) were randomly selected for model evaluation (Figure S4). Training was performed using a combination of $PM_{2.5}$ components, meteorological, and emission features. The model showed a prediction $R^2$ of 0.68 and RMSE of 17.26 µg/m$^3$."

2. Poor R2 is not only due to the features also to the algorithm itself and the hyperparameters. How to determine what exactly is the cause? R2 could have been examined after the authors conducted the identical investigation using linear regression, what then is the purpose of LightGBM? Does not this simply because LightGBM produces superior outcomes in comparison to linear regression? And how did the authors ensure the current LightGBM-based mode is better than other models?

- Thanks for the comment. It is indeed difficult to fully distinguish what causes the low $R^2$. We designed a complementary experiment to measure the impact of the model itself by comparing popular regression models (including Linear Polynomial Regression, Quadratic Polynomial Regression, Random Forest, XGBoost, and LightGBM) with the same features ($PM_{2.5}$ components). The results (Table A6) show that all models show similar performance, e.g., all models show lower $R^2$ in the winter in the BTH (0.16 - 0.4), and higher $R^2$ in the SCB region (0.7 - 0.8). This is a side evidence that the low $R^2$ is more affected by the features than the model itself, as the commonly used regression models can hardly obtain high $R^2$ with insufficient

explanatory features (e.g., chemical component features in winter in BTH).

LightGBM is used because it can better capture the non-linear relationship between the input features and the target, compensating for the shortcomings of linear models. Linear models can only capture the effects of some linear processes on $PM_{2.5}$ concentrations, e.g. more primary emissions lead to higher $PM_{2.5}$ concentrations when secondary pollution is low. However, the effects of emissions and meteorological factors on $PM_{2.5}$ concentrations are highly non-linear in scenarios with high secondary pollution, for example, high relative humidity increases the total $PM_{2.5}$ concentration by promoting the hygroscopic growth of $PM_{2.5}$ and the production of secondary particulate matter on the one hand, but on the other hand, it promotes the deposition of particulate matter, which reduces the concentration of $PM_{2.5}$. The non-linear models such as LightGBM can better describe the nonlinear process among meteorological-emission-$PM_{2.5}$ concentration and further identify the sources of model bias in CTMs. The model comparison results (Table A6) also show the better prediction ability of the nonlinear models. The LightGBM model shows superior accuracy and robustness than the other models. Therefore, we chose the LightGBM model to identify the source of simulation bias for CTMs. We have added the discussion in Section 3.3.

**Changes in Lines 249-258:** "I In addition, the main objective of this study was diagnosing the contributors to CMAQ simulation biases using machine learning, rather than for prediction. Since meteorology or emissions can only partially explain the simulation bias, a low $R^2$ is justified when fitting the model with only meteorology or emissions variables (Figure 4). We designed a complementary experiment to measure the impact of the model itself by comparing popular regression models (including multiple linear regression, polynomial regression (degree:2), Random Forest, XGBoost, and LightGBM) with the same features ($PM_{2.5}$ components). The results (Table A6) show that all models show similar performance, e.g., all models show lower $R^2$ in the winter in the BTH (0.16 - 0.4), and higher $R^2$ in the SCB region (0.7 - 0.8). This is a side evidence that the low $R^2$ is more affected by the features than the model itself, as the commonly used regression models can hardly

obtain high $R^2$ with insufficient explanatory features (e.g., chemical component features in winter in BTH)."

3. Why not develop distinct models for each chemical component individually?

 - Thanks for the comment. The main reason is that the observation data of chemical components are not openly available in China. The observation network of chemical components has been set up in China, but the data are closed-source. Currently, there are open-source machine learning based chemical reanalysis datasets in China, like (Liu et al., 2022; Wei et al., 2023), however, due to their high uncertainty, we cannot use them as the true values to build models. We attempted to crawl the data from the literature, but the quantity and quality of the data was insufficient. We hope to make China's air quality observation data more open source in the future. Using sufficient observed data on chemical composition and combining it with machine learning models, the sources of bias in CTM can be more accurately identified to guide model improvement.

4. Why not separate each chemical components bias into its own model ?

 - Thanks for the comment. As mentioned above, the main reason that we do not model each chemical bias separately is that there are no publicly available observations of chemical composition in China. In the future, we hope to strengthen cooperation and promote data sharing to more accurately identify CTMs simulation deviations and guide CTMs' improvement.

5. What effect do feature interactions have?

 - Thanks for the comment. For LightGBM, the interaction between features (multicollinearity) does not affect model predictive power. LightGBM uses a Leaf-wise Tree Growth algorithm, a node-splitting strategy that is less affected by covariance (Ke et al., 2017). The most extreme case of multicollinearity is when there are two identical features. When one feature is used, the decision tree will not use another feature because it adds no new valid information. The multicollinearity

between features will affect features' relative importance. If two variables are correlated, the importance of both will slightly decrease. Previous studies (Hou et al., 2022; Ye et al., 2022) have used ML to explain the causes of air pollution and model bias, and although there was multicollinearity between the input features they used, they got reliable conclusions, showing the slight influence of multicollinearity and the reliability of tree-based machine learning methods.

**Changes in Lines 244-249:** "Although we filtered the features according to their relative importance and priori knowledge, collinearity still exists among the input features. Features collinearity does not affect the performance of tree-based models like Random Forest and LightGBM (Belgiu and Drăguț, 2016; Chen et al., 2016; Ke et al., 2017), but the contribution of a single feature might be slightly influenced by other features. Previous studies (Hou et al., 2022; Ye et al., 2022) have used ML to explain the causes of air pollution and model bias, and although there was multicollinearity between the input features they used, they got reliable conclusions, showing the slight influence of multicollinearity and the reliability of tree-based machine learning methods."

**Comments from Reviewer #2:**

**General comments:** This study uses ML algorithms to determine the source of CTM bias, it is an interesting and innovative study and the approach has the potential to be generalised to similar studies. After revisions, the manuscript has been greatly improved and here are a few minor issues that need to be addressed.

- Thank you for your feedback and for taking the time to review our manuscript. We carefully address the concern and provide a detailed response to each of your comments in the revision.

**Specific comments:**

1. It is suggested to add the description of observation data, because this study is based

on it, including the number and distribution of stations, and the number of effective observation dramas.

- Thanks for the comment. We have added the description of observation data to give more information.

**Changes in Lines 65-67:** "A total of about 350,000 valid observations were selected. The distribution of observation sites (about 1200) is shown in Figure S1. The stations are unevenly distributed, with dense stations in eastern populated areas and sparse stations in western Xinjiang and Tibet."

2. In the abstract, please show important results of model performance.

- Thanks for the comment. We have added the model performance results in abstract. The ML model can capture the complex relationship between input variables and simulation bias well (test $R^2$ = 0.68). Small performance gap between training and testing indicated model's good generalization ability (delta $R^2$: 0.16 – 0.18).

**Changes in Lines 19-21:** "The ML model can capture the complex relationship between input variables and simulation bias well (test $R^2$ = 0.68) with small performance gap between training and validation (delta $R^2$: 0.16 – 0.18)."

3. L45 Add full name of XGboost.

- Thank you for your comment. We have added the full name of XGboost, as "eXtreme Gradient Boosting", which is an optimised implementation of Gradient Boosting Decision Trees that improves speed and performance.

4. L54: "lightGBM" or LightGBM? Please make it case-sensitive.

- Thanks for the comment. We uniformly modified the expression: 'LightGBM', and scrutinised the whole manuscript

5. L111: Add formula of R2 and RMSE or reference.

   - Thanks for the comment. We added the formula of $R^2$ and RMSE in Section 2.3.

**Changes in Lines 111-115:** "The dataset was randomly divided into five parts, one was taken in turn as a test and the rest was used for training, which was repeated five times, and then the mean coefficient of determination ($R^2$) and the root mean square error (RMSE) were calculated (Wei et al., 2020).

$$R^2 = 1 - \frac{\sum_i (y_i - f_i)^2}{\sum_i (y_i - \hat{y})^2} \tag{1}$$

$$\text{RMSE} = \sqrt{\frac{1}{n} \sum_{i=1}^{n} (y_i - f_i)^2} \tag{2}$$

"

6. L153: "an" – a.

   - Thanks for the comment. We apologise for our carelessness, we have corrected the expression and carefully checked the entire manuscript for grammatical correctness.

Reference

Belgiu, M. and Drăguţ, L.: Random forest in remote sensing: A review of applications and future directions, ISPRS-J. Photogramm. Remote Sens., 114, 24-31, 2016.

Chen, T. Q., Guestrin, C., and Assoc Comp, M.: XGBoost: A Scalable Tree Boosting System, 22nd ACM SIGKDD International Conference on Knowledge Discovery and Data Mining (KDD), San Francisco, CA, Aug 13-17, WOS:000485529800092, 785-794, 10.1145/2939672.2939785, 2016.

Hou, L. L., Dai, Q. L., Song, C. B., Liu, B. W., Guo, F. Z., Dai, T. J., Li, L. X., Liu, B. S., Bi, X. H., Zhang, Y. F., and Feng, Y. C.: Revealing Drivers of Haze Pollution by Explainable Machine Learning, Environmental Science & Technology Letters, 9, 112-119, 10.1021/acs.estlett.1c00865, 2022.

Ke, G., Meng, Q., Finley, T., Wang, T., Chen, W., Ma, W., Ye, Q., and Liu, T.-Y.: Lightgbm: A highly efficient gradient boosting decision tree, Advances in neural information processing systems, 30, 2017.

Liu, S., Geng, G., Xiao, Q., Zheng, Y., Liu, X., Cheng, J., and Zhang, Q.: Tracking Daily Concentrations of PM2.5 Chemical Composition in China since 2000, Environ. Sci. Technol., 56, 16517-16527, 10.1021/acs.est.2c06510, 2022.

Wei, J., Li, Z. Q., Cribb, M., Huang, W., Xue, W. H., Sun, L., Guo, J. P., Peng, Y. R., Li, J., Lyapustin, A., Liu, L., Wu, H., and Song, Y. M.: Improved 1 km resolution PM2.5 estimates across China using enhanced space-time extremely randomized trees, Atmos. Chem. Phys., 20, 3273-3289, 10.5194/acp-20-3273-2020, 2020.

Wei, J., Li, Z., Chen, X., Li, C., Sun, Y., Wang, J., Lyapustin, A., Brasseur, G. P., Jiang, M., Sun, L., Wang, T., Jung, C. H., Qiu, B., Fang, C., Liu, X., Hao, J., Wang, Y., Zhan, M., Song, X., and Liu, Y.: Separating Daily 1 km PM2.5 Inorganic Chemical Composition in China since 2000 via Deep Learning Integrating

Ground, Satellite, and Model Data, Environ. Sci. Technol., 57, 18282-18295, 10.1021/acs.est.3c00272, 2023.

Ye, X., Wang, X., and Zhang, L.: Diagnosing the Model Bias in Simulating Daily Surface Ozone Variability Using a Machine Learning Method: The Effects of Dry Deposition and Cloud Optical Depth, Environ. Sci. Technol., 56, 16665-16675, 10.1021/acs.est.2c05712, 2022.

---

## Author Response (AR3)

**Point-by-point responses to the comments & suggestions from the editor**

**Journal: Geoscientific Model Development**

**Manuscript ID: EGUSPHERE-2023-1531**

**Title: "Diagnosing drivers of PM$_{2.5}$ simulation biases in China from meteorology, chemical composition, and emission sources using an efficient machine learning method"**

**Comments from the editor:**

Please reply to the reviewers' comments carefully.

- We appreciate the editor's efforts on this manuscript. In the revision, we carefully revised the manuscript based on reviewers' comments and made detailed responses. We hope to meet the requirements of the journal. Please contact us if there are any problems.

**Comments from Reviewer #1:**

Summary: In this manuscript, the author introduces LightGBM, a tree-based regression method, as a powerful tool for evaluating the performance of the Community Multiscale Air Quality (CMAQ) model. The primary focus is on diagnosing the CMAQ's effectiveness in pinpointing the predominant contributing factors responsible for prediction bias, particularly in relation to the prediction of PM 2.5 concentration. To comprehensively assess potential biases associated with each source, LightGBM is employed to conduct separate time series regressions for features grouped into three major sources

**Major comments:**

After reading the manuscript, I think some major comments from Anonymous Reviewer #1 of last round are still not adequately addressed. In my opinion, the author should clearly address the following aspects:

- Thank you for your feedback and time to review our manuscript. We apologize for not adequately addressing the comments in the last revision. We have made careful revisions and have provided detailed responses to each of your comments in this revision. Thank you again for your time and consideration. We look forward to

working closely with you to address your concerns and make the necessary improvements.

1. Dataset setting:

a. Provide a clear description of the 350,000 valid observations, specifying whether it represents the sum of all time series data points across multiple monitoring stations. Clearly state the methodology for training and testing data separation.

    - Thanks for the comment. The observation data is the sum of all time series data points across multiple monitoring stations. We have added the specific description in Section 2.1. The training and testing data were randomly separated in an 8:2 ratio for all stations in the region of interest. We have clarified it in Section 2.3.

     **Changes in Lines 65-66:** "A total of about 350,000 observations meeting quality control criteria were selected from all time series data points across multiple monitoring stations."

     **Changes in Lines 118-119:** "The separate test sets (not involved in the training and CV hyperparameter selection process) were divided by randomly sampling 20% of the data from all stations in the region of interest."

b. Clarify how random samples of observations are selected. Specify whether the 20% random sampling is performed at the station level or across all stations in the region of interest.

    - Thanks for the comment. The 20% random sampling was performed across all stations in the region of interest. We have clarified it in Section 2.3.

     **Changes in Lines 118-119:** "The separate test sets (not involved in the training and CV hyperparameter selection process) were divided by randomly sampling 20% of the data from all stations in the region of interest."

c. Time series data usually cannot be directly learned through tree-based model without additional pre-processing/feature engineering. Discuss the absence of data preparation and feature engineering before feeding data into tree-based models. If temporal structure is considered negligible, provide justification; otherwise, explain the approach taken to handle temporal aspects.

- Thanks for the comment. We combined time series data from multiple stations, eliminated the extreme values of 0.1%, and did not specifically preprocess the temporal structure. In previous studies of pollution prediction based on tree models, time-related features are usually added to characterize the temporal pattern of pollutant changes to further improve the prediction ability, e.g., Wei et al. (2021a) improved the model performance by adding temporal features of day of year and Unix timestamps. However, the goal of this study is to identify the contributors to the simulation bias based on feature importance rather than prediction, and the inclusion of temporal features cannot provide any useful information for us instead it is difficult to attribute them to meteorological or emissions contributions. For example, the simulated bias shows a clear temporal pattern, being larger in the winter and smaller in the summer, so temporal features would show a high contribution to the simulation bias, but provide no valid information. Therefore, we did not include temporal features in our model.

**Changes in Lines 268-272:** "Previous pollution prediction studies based on tree models usually added the time-related features to describe the temporal pattern of pollutant changes to further improve the prediction ability, e.g., Wei et al. (2021a) improved the model performance by adding temporal features of day of year and Unix timestamps. However, the inclusion of temporal features cannot provide any useful information about contributors of simulation biases instead it is difficult to attribute them to meteorological or emissions contributions.Therefore, temporal features were not included in our model."

2. Tree-based model justification
a. In the section (L95-100), provide examples of similar applications in terms of dataset, model, and research area. Demonstrate why tree-based methods are suitable for the specific dataset. Justify the selection of tree-based models beyond considerations of memory and computation time.

- Thanks for the comment. We have provided examples of similar applications. Tree-based ML models typically outperform deep learning approaches in tabular data due to limited data number and relative sample structure (compared to image, video, and natural language). The LightGBM model has shown accurate performance in

many fields, with fast speeds, less overfitting, and independence from collinearity. We have justified the selection of tree-based models in Section 2.3.

**Changes in Lines 95-109:** "Tree-based ML models typically outperform deep learning approaches in tabular data (e.g., air pollutant observation datasets), and thus have been widely developed (Grinsztajn et al., 2022). Wei et al. (2021a) compared several models when reconstructing $PM_{2.5}$ data records in China and found that the tree model showed superior performance. The LightGBM model is an optimized Gradient Boosting Decision Tree (GBDT) (Ke et al., 2017), and has shown accurate performance in many fields (Wei et al., 2021b; Yan et al., 2021; Sun et al., 2020; Liang et al., 2020). Compared to XGBoost, a widely used GBDT, LightGBM uses Histogram's decision tree algorithm along with Gradient-based One-Side Sampling (GOSS), which saves memory and computation time (Ke et al., 2017). Three tree-based models, Random Forest, XGBoost, and LightGBM, were compared in our previous study (Wang et al., 2023). Using the same input data and hyperparameters, LightGBM is as accurate as XGBoost, but faster and less overfitting (the difference in accuracy between training and testing). Besides, Multiple colinearities between features such as pollutant concentrations and meteorological factors can greatly affect the performance of traditional linear models. When independent variables are correlated, changes in one variable are associated with changes in the other, making it difficult for the model to independently estimate the relationship between each independent and dependent variable. However these collinearities does not affect the performance of tree-based models like Random Forest and LightGBM, because they do not require the assumption of feature independence (Belgiu and Drăguţ, 2016; Chen et al., 2016; Ke et al., 2017). So the lightGBM model was used to diagnose $PM_{2.5}$ simulation biases in this study."

b. Introduce a discussion on multicollinearity in the methodology section.

   - Thanks for the comment. We have added the discussion of multicollinearity in Section 2.3.

   **Changes in Lines 103-109:** "Besides, Multiple colinearities between features

such as pollutant concentrations and meteorological factors can greatly affect the performance of traditional linear models. When independent variables are correlated, changes in one variable are associated with changes in the other, making it difficult for the model to independently estimate the relationship between each independent and dependent variable. However, these collinearities do not affect the performance of tree-based models like Random Forest and LightGBM, because they do not require the assumption of feature independence (Belgiu and Drăguţ, 2016; Chen et al., 2016; Ke et al., 2017)."

3. Cross validation

a. Clearly explain how cross-validation is performed and provide a statement on how the two metrics (R^2 and RMSE) influence model selection decisions. Clearly articulate the criteria for jointly considering model performance using these two metrics.

 - Thanks for the comment. Cross-validation (5-fold) combined with RMSE was used to select hyperparameters. Two metrics ($R^2$ and RMSE) were used to evaluate the model performance in the separate test sets. Higher $R^2$ and lower RMSE represent better model performance. We have clarified cross-validation and model evaluation in Section 2.3.

 **Changes in Lines 114-119:** "Cross-validation (5-fold) combined with RMSE was used to select hyperparameters. The dataset was randomly divided into five parts, one was taken in turn as the test set and the rest was used for training, which was repeated five times and the average test RMSE was calculated. Looping to increase model complexity, ending the loop and returning to the hyperparameters when the model test RMSE does not decrease significantly (< 0.01) or the gap between training and test RMSE increases significantly (< 0.05). The separate test sets (not involved in the training and CV hyperparameter selection process) were divided by randomly sampling 20% of the data from all stations in the region of interest."

 **Changes in Lines 109-111:** "Two metrics were calculated to evaluate model performance, including the coefficient of determination ($R^2$) and the root mean square error (RMSE) (Wei et al., 2020)."

Minor comments:

L15. Clarify the term "efficient" to provide a precise understanding within the context of this study.

    - Thanks for the comment. We defined the "efficient" as "fast speed and low requirement of computational resources", and have reorganized the corresponding expression.

    **Changes in Lines 14-18:** "Accurate diagnosis of simulation biases is critical for improvement of models, interpretation of results, and management of air quality, especially for the simulation of fine particulate matter ($PM_{2.5}$). In this study, an efficient method with fast speed and low requirement of computational resources based on tree-based machine learning (ML) method, the Light Gradient Boosting Machine (LightGBM), was designed to diagnose CTMs simulation biases."

L16. Instead of broadly referring to "machine learning," explicitly specify that LightGBM is a tree-based method. Additionally, consider breaking the sentence into two for enhanced readability.

    - Thanks for the comment. We changed the "machine learning" to "tree-based machine learning (ML) method, the Light Gradient Boosting Machine (LightGBM)". We split the sentence in two and reorganized the expression accordingly.

    **Changes in Lines 16-20:** "In this study, an efficient method with fast speed and low requirement of computational resources based on tree-based machine learning (ML) method, the Light Gradient Boosting Machine (LightGBM), was designed to diagnose CTMs simulation biases. The drivers of the Community Multiscale Air Quality (CMAQ) model biases compared to observations in simulating $PM_{2.5}$ concentrations from three perspectives of meteorology, chemical composition, and emission sources."

L20. Reevaluate the assertion that an R^2 value of 0.68 constitutes good performance. Provide references from existing literature to substantiate this claim. Additionally, the relative performance gap of 0.16 is about 23.5% of 0.68, which might not be

compelling enough; its significance in the context of overfitting and the ability to be applied to other fields is weak.

- Thanks for the comment. We have reevaluated the model. The ML models were separately trained by meteorology, $PM_{2.5}$ components, and source sectors for different regions and seasons, and the test sets were used to measure the model performance (Figure 4). The meteorology, $PM_{2.5}$ components, and source sectors can partially explain the simulation bias, with mean test $R^2$ of 0.64, 0.52, and 0.50, respectively, and the RMSE was 17.41, 19.82, and 19.56 µg/m³, respectively. We removed inappropriate content and have changed the description of the overfitting issue.

**Changes in Lines 167-172:** "The ML models were separately trained by meteorology, $PM_{2.5}$ components, and source sectors for different regions and seasons, and separate test sets were used to measure the model performance (Figure 4). All three feature combinations can partially explain the simulation bias. The mean test $R^2$ for meteorology, $PM_{2.5}$ components, and source sectors were 0.64, 0.52, and 0.50, respectively, and the RMSE were 17.41, 19.82, and 19.56 µg/m³, respectively. The model performed better in summer than in winter. This may be attributed to the high simulation biases in winter due to severe $PM_{2.5}$ pollution and complex causes, while $PM_{2.5}$ pollution in summer is lighter with lower CMAQ simulation bias."

**Changes in Lines 266-267:** "Besides, LightGBM shows comparable accuracy to XGBoost but is faster and shows smaller accuracy gaps between training and testing with less overfitting."

L65. Revise the description of "valid" observations to explicitly convey that these observations adhere to the quality control criteria outlined in L62-64. Reorganize the sentences for better coherence.

- Thanks for the comment. We have revised the description of "valid" to "meeting quality control criteria", and reorganized the sentences for better coherence.

**Changes in Lines 65-66:** "A total of about 350,000 observations meeting quality control criteria were selected from all time series data points across multiple monitoring stations."

L76. Consider either elaborating on the model's enhancements or removing the sentence for conciseness.

- Thanks for the suggestion. We took your advice and removed the sentences for conciseness.

L80-82. Clearly indicate that CMAQ is employed for simulating PM 2.5 components when introducing the model. Adjust the sequence of information to improve logical flow.

- Thanks for the comment. We have adjust the sequence of the CMAQ introduction for better consistency.

**Changes in Lines 72-73:** "The CMAQ simulation (36 km×36 km) was carried out to simulate $PM_{2.5}$ components in mainland China and surrounding regions in 2019. The list of $PM_{2.5}$ components simulated by CMAQ is shown in Table A1."

L86. Enhance the fluency by adding a connecting word at the beginning of the sentence.

- Thanks for the comment. We have reorganized the presentation to make the sentences more coherent and checked the coherence of the entire manuscript.

**Changes in Lines 85-86:** "PPM from different source sectors are tracked by non-reactive tracers ($10^{-5}$ of the PPM emission rates), and source-specific PPM concentrations are then calculated by multiplying the tracer with $10^5$."

L119-120. Define "success" in quantitative or qualitative terms to provide a clearer understanding of the criteria for evaluating success.

- Thanks for the comment. We eliminated the vague expression "success" and quantified it with specific statistics.

**Changes in Lines 132-134:** "Observed $PM_{2.5}$ concentrations were highest in BTH (51.172 μg/m$^3$) and lowest in PRD (28.273 μg/m$^3$). The CMAQ model underestimates $PM_{2.5}$ concentrations of -8.59 μg/m$^3$, -2.66 μg/m$^3$, -6.21 μg/m$^3$, and -19.25 μg/m$^3$ in the BTH, YRD, PRD, and NWCHN, respectively (Figure 1b)."

L170. Remove the extra period before the citation.

- Thanks for the comment. We have removed the extra period before the citation, and we apologize for our carelessness and have carefully examined the entire manuscript.

L245. Replace "Features collinearity" with "Multicollinearity among features."

    - Thanks for the comment. We have replaced "Features collinearity" with "Multicollinearity among features."

    **Changes in Lines 252-253:** "Multicollinearity among features does not affect the performance of tree-based models like Random Forest and LightGBM"

Table A6. Use bold font to highlight the best metric performance. Additionally, if XGB and LGB exhibit similar performance, with XGB slightly superior, consider including computational time as an additional metric to justify the preference for LGB over XGB.

    - Thanks for the comment. We have highlighted the best metric performance and include computational time as an additional metric in Table A6.

    **Changes in Lines 266-267:** "Besides, LightGBM shows comparable accuracy to XGBoost, but is faster and shows smaller accuracy gaps between training and testing with less overfitting."

**Comments from Reviewer #2:**

This manuscript requires additional revisions.

1. Cross-validation is mainly used for hyperparameter tuning, to select the best hyperparameter combination and prevent overfitting to the training data. After selecting the final model, we still need to evaluate performance on an independent test set to check the model's ability to generalize to real data. However, I noticed the evaluation of model performance in this paper is still based on 5-fold cross-validation (e.g., Lines 110-111 and Lines 158-159). The authors should rewrite Section 3.2 by using the independent testing data for model evaluation instead of cross-validation.

    - Thanks for the comment. We have rewritten the first part of Section 3.2 by using the independent testing data for model evaluation. Specifically, The ML models were separately trained by meteorology, $PM_{2.5}$ components, and source sectors for different regions and seasons, and separate test sets were used to measure the model

performance (Figure 4). The 5-fold cross-validation combined with RMSE was used to select hyperparameters.

**Changes in Lines 167-172:** "The ML models were separately trained by meteorology, PM$_{2.5}$ components, and source sectors for different regions and seasons, and separate test sets were used to measure the model performance (Figure 4). All three feature combinations can partially explain the simulation bias. The mean test R$^2$ for meteorology, PM$_{2.5}$ components, and source sectors were 0.64, 0.52, and 0.50, respectively, and the RMSE was 17.41, 19.82, and 19.56 μg/m$^3$, respectively. The model performed better in summer than in winter. This may be attributed to the high simulation biases in winter due to severe PM$_{2.5}$ pollution and complex causes, while PM$_{2.5}$ pollution in summer is lighter with lower CMAQ simulation bias."

**Changes in Lines 114-119:** "Cross-validation (5-fold) combine with RMSE to select hyperparameters. The dataset was randomly divided into five parts, one was taken in turn as the test set and the rest was used for training, which was repeated five times and the average test RMSE was calculated. Looping to increase model complexity, ending the loop and returning to the hyperparameters when the model test RMSE does not decrease significantly ($< 0.01$) or the gap between training and test RMSE increases significantly ($< 0.05$). The separate test sets (not involved in the training and CV hyperparameter selection process) were divided by randomly sampling 20% of the data from all stations in the region of interest."

2. LightGBM has two types of feature importance, namely "split" and "gain." Could you please clarify which feature importance type was used for analysis in this study? I think clearly mentioning the type used would strengthen the analysis.

- Thanks for the comment. We used "gain" to measure feature importance, which is the size of the gain resulting from splitting through a certain feature. The type of "split" is the number of splits using a particular feature. For some highly indicative categorical features that may split only once during tree growth, but have high importance, at which time the split method may be inaccurate. So we used "gain" to measure feature importance.

**Changes in Lines 120-122:** "The target variable was set to be the difference between observed and simulated daily PM$_{2.5}$ concentrations, and the key contributors

to the simulation bias were then determined by the relative importance (calculated by gain) of the input features (Ye et al., 2022; Loyola-González et al., 2023)"

**Reference**

Belgiu, M. and Drăguţ, L.: Random forest in remote sensing: A review of applications and future directions, ISPRS-J. Photogramm. Remote Sens., 114, 24-31, 2016.

Chen, T. Q., Guestrin, C., and Assoc Comp, M.: XGBoost: A Scalable Tree Boosting System, 22nd ACM SIGKDD International Conference on Knowledge Discovery and Data Mining (KDD), San Francisco, CA, Aug 13-17, WOS:000485529800092, 785-794,  10.1145/2939672.2939785, 2016.

Grinsztajn, L., Oyallon, E., and Varoquaux, G.: Why do tree-based models still outperform deep learning on tabular data?, arXiv preprint arXiv:2207.08815, 2022.

Ke, G., Meng, Q., Finley, T., Wang, T., Chen, W., Ma, W., Ye, Q., and Liu, T.-Y.: Lightgbm: A highly efficient gradient boosting decision tree, Advances in neural information processing systems, 30, 2017.

Liang, W., Luo, S., Zhao, G., and Wu, H.: Predicting hard rock pillar stability using GBDT, XGBoost, and LightGBM algorithms, Mathematics, 8, 765, 2020.

Loyola-González, O., Ramírez-Sáyago, E., and Medina-Pérez, M. A.: Towards improving decision tree induction by combining split evaluation measures, Knowledge-Based Systems, 277, 110832, https://doi.org/10.1016/j.knosys.2023.110832, 2023.

Sun, X., Liu, M., and Sima, Z.: A novel cryptocurrency price trend forecasting model based on LightGBM, Finance Research Letters, 32, 101084, 2020.

Wang, S., Wang, P., Zhang, R., Meng, X., Kan, H., and Zhang, H.: Estimating particulate matter concentrations and meteorological contributions in China during 2000–2020, Chemosphere, 330, 138742, https://doi.org/10.1016/j.chemosphere.2023.138742, 2023.

Wei, J., Li, Z., Lyapustin, A., Sun, L., Peng, Y., Xue, W., Su, T., and Cribb, M.: Reconstructing 1-km-resolution high-quality PM2. 5 data records from 2000 to 2018 in China: spatiotemporal variations and policy implications, Remote Sensing of Environment, 252, 112136, 2021a.

Wei, J., Li, Z., Pinker, R. T., Wang, J., Sun, L., Xue, W., Li, R., and Cribb, M.: Himawari-8-derived diurnal variations in ground-level PM2.5 pollution across China using the fast space-time Light Gradient Boosting Machine (LightGBM), Atmos. Chem. Phys., 21, 7863-7880, 10.5194/acp-21-7863-2021, 2021b.

Wei, J., Li, Z. Q., Cribb, M., Huang, W., Xue, W. H., Sun, L., Guo, J. P., Peng, Y. R., Li, J., Lyapustin, A., Liu, L., Wu, H., and Song, Y. M.: Improved 1 km resolution PM2.5 estimates across China using enhanced space-time extremely randomized trees, Atmos. Chem. Phys., 20, 3273-3289, 10.5194/acp-20-3273-2020, 2020.

Yan, J., Xu, Y., Cheng, Q., Jiang, S., Wang, Q., Xiao, Y., Ma, C., Yan, J., and Wang, X.: LightGBM: accelerated genomically designed crop breeding through ensemble learning, Genome Biology, 22, 1-24, 2021.

Ye, X., Wang, X., and Zhang, L.: Diagnosing the Model Bias in Simulating Daily Surface Ozone Variability Using a Machine Learning Method: The Effects of Dry Deposition and Cloud Optical Depth, Environ. Sci. Technol., 56, 16665-16675, 10.1021/acs.est.2c05712, 2022.

---

## Author Response (AR4)

**Point-by-point responses to the comments & suggestions from the editor**

**Journal: Geoscientific Model Development**

**Manuscript ID: EGUSPHERE-2023-1531**

**Title: "Diagnosing drivers of PM$_{2.5}$ simulation biases in China from meteorology, chemical composition, and emission sources using an efficient machine learning method"**

**Comments from the editor:**

Please revise the manuscript and finalize it, as suggested by the reviewers.

- We appreciate the editor's efforts on this manuscript. In the revision, we carefully revised the manuscript, double-checked the expression and grammar, and made detailed responses. We hope to meet the requirements of the journal. Please contact us if there are any problems.

L31. "particulate matters"

**Change to:** "particulate matter"

L53-54. "Moreover, due to the strong influence of emissions, it is instructive to diagnose CTMs biases of PM$_{2.5}$ based on a source-appointment perspective."

**Change to:** "Moreover, given the significant impact of emissions, it is instructive to diagnose CTMs biases of PM$_{2.5}$ based on a source-appointment perspective."

L60-61. "This study focused on 2019 because of the large number of observations, the reliability of the emission inventories, and without interference of COVID19."

**Change to:** "This study specifically targets the year of 2019 due to the extensive availability of observational data, the reliability of emission inventories, and the absence of COVID-19-related disruptions."

L65-66. "A total of about 350,000 observations meeting quality control criteria were selected from all time series data points across multiple monitoring stations."

**Change to:** "Approximately 350,000 observations, which met the quality control criteria, were selected from the entire time series data points collected from various monitoring stations."

L126-128. "Seven sectoral sources (industry, energy, residential, transportation, agriculture, open burning, and biogenic emission) were used to quantify the contribution to the simulation bias."

**Change to:** "The contributions to the simulation bias were quantified using seven sectoral sources: industry, energy, residential, transportation, agriculture, open burning, and biogenic emissions."